# Cell-intrinsic and -extrinsic mechanisms promote cell-type-specific cytokinetic diversity

Tim Davies[1], Han X Kim[1,2], Natalia Romano Spica[1], Benjamin J Lesea-Pringle[1], Julien Dumont[3], Mimi Shirasu-Hiza[2], Julie C Canman[1]*

[1]Department of Pathology and Cell Biology, Columbia University Medical Center, New York, United States; [2]Department of Genetics and Development, Columbia University Medical Center, New York, United States; [3]Institut Jacques Monod, CNRS UMR 7592, Université Paris Diderot, Paris, France

**Abstract** Cytokinesis, the physical division of one cell into two, is powered by constriction of an actomyosin contractile ring. It has long been assumed that all animal cells divide by a similar molecular mechanism, but growing evidence suggests that cytokinetic regulation in individual cell types has more variation than previously realized. In the four-cell *Caenorhabditis elegans* embryo, each blastomere has a distinct cell fate, specified by conserved pathways. Using fast-acting temperature-sensitive mutants and acute drug treatment, we identified cell-type-specific variation in the cytokinetic requirement for a robust formin$^{CYK-1}$-dependent filamentous-actin (F-actin) cytoskeleton. In one cell (P2), this cytokinetic variation is cell-intrinsically regulated, whereas in another cell (EMS) this variation is cell-extrinsically regulated, dependent on both Src$^{SRC-1}$ signaling and direct contact with its neighbor cell, P2. Thus, both cell-intrinsic and -extrinsic mechanisms control cytokinetic variation in individual cell types and can protect against division failure when the contractile ring is weakened.

DOI: https://doi.org/10.7554/eLife.36204.001

*For correspondence: jcanman@gmail.com

Competing interests: The authors declare that no competing interests exist.

## Introduction

Cytokinesis is the physical division of one cell into two daughter cells, which occurs at the end of the cell cycle. In animal cells, cytokinesis is driven by the equatorial constriction of an actomyosin contractile ring, composed of diaphanous family formin-nucleated F-actin and the motor myosin-II (for review see [*Cheffings et al., 2016*; *D'Avino et al., 2015*; *Green et al., 2012*; *Mandato et al., 2000*; *Mishima, 2016*; *Pollard, 2010*]). Cytokinesis failure, which results in a binucleate tetraploid (polyploid) cell, can lead to human diseases including blood syndromes, neurological disorders, and cancer (*Bione et al., 1998*; *Moulding et al., 2007*; *Dieterich et al., 2009*; *Vinciguerra et al., 2010*; *Lacroix and Maddox, 2012*; *Iolascon et al., 2013*; *Liljeholm et al., 2013*; *Ferrer et al., 2014*; *Ganem et al., 2014*, *2007*; *Tormos et al., 2015*). While it has long been assumed that all animal cells divide by a similar molecular mechanism, it is becoming increasingly clear that the functional regulation of cytokinesis has more diversity, or variation in mechanistic and regulatory pathways, than previously appreciated (*Herszterg et al., 2014*; *Guillot and Lecuit, 2013*; *Founounou et al., 2013*; *Herszterg et al., 2013*; *De Santis Puzzonia et al., 2016*; *Choudhary et al., 2013*; *Wheatley et al., 1997*; *Stopp et al., 2017*). In many animals (including humans), specific cell types or cell lineages within the organism are programmed to fail in cytokinesis and become bi- or multi-nucleate (e.g. osteoclasts in bone, megakaryocytes in blood, cardiomyocytes in the heart, hepatocytes in the liver) (*Lacroix and Maddox, 2012*; *Tormos et al., 2015*; *Zimmet and Ravid, 2000*; *Duncan, 2013*; *Takegahara et al., 2016*). Thus, in some cell types, cytokinesis failure occurs normally

**eLife digest** The successful division of one cell into two is essential for all organisms to live, grow and reproduce. For an animal cell, the nucleus – the compartment containing the genetic material – must divide before the surrounding material. The rest of the cell, called the cytoplasm, physically separates later in a process known as cytokinesis.

Cytokinesis in animal cells is driven by the formation of a ring in the middle of the dividing cell. The ring is composed of myosin motor proteins and filaments made of a protein called actin. The movements of the motor proteins along the filaments cause the ring to contract and tighten. This pulls the cell membrane inward and physically pinches the cell into two. For a long time, the mechanism of cytokinesis was assumed to be same across different types of animal cell, but later evidence suggested otherwise. For example, in liver, heat and bone cells, cytokinesis naturally fails during development to create cells with two or more nuclei. If a similar 'failure' happened in other cell types, it could lead to diseases such as cancers or blood disorders. This raised the question: what are the molecular mechanisms that allow cytokinesis to happen differently in different cell types?

Davies et al. investigated this question using embryos of the worm *Caenorhabditis elegans* at a stage in their development when they consist of just four cells. The proteins forming the contractile ring in this worm are the same as those in humans. However, in the worm, the contractile ring can easily be damaged using chemical inhibitors or by mutating the genes that encode its proteins.

Davies et al. show that when the contractile ring was damaged, two of the four cells in the worm embryo still divided successfully. This result indicates the existence of new mechanisms to divide the cytoplasm that allow division even with a weak contractile ring. In a further experiment, the embryos were dissected to isolate each of the four cells. Davies et al. saw that one of the two dividing cells could still divide on its own, while the other cell could not. This shows that this new method of cytokinesis is regulated both by factors inherent to the dividing cell and by external signals from other cells. Moreover, one of these extrinsic signals was found to be a signaling protein that had previously been implicated in human cancers.

Future work will determine if these variations in cytokinesis between the different cell types found in the worm apply to humans too; and, more importantly from a therapeutic standpoint, if these new mechanisms exist in human cancers.

DOI: https://doi.org/10.7554/eLife.36204.002

during development and/or homeostasis and, in other cell types, cytokinesis failure can be pathogenic (*Lacroix and Maddox, 2012*; *Tormos et al., 2015*; *Zimmet and Ravid, 2000*; *Duncan, 2013*; *Takegahara et al., 2016*), indicating a high degree of cellular variation in both the regulation of cytokinesis and the consequences of cytokinesis failure.

As further support for cell-type-specific cytokinetic variation, genomic analysis has revealed that organism-wide mutations in cytokinesis genes are associated with cell-type-specific disruption of cell division in flies, fish, worms, rodents, and even humans (*Bione et al., 1998*; *Moulding et al., 2007*; *Vinciguerra et al., 2010*; *Liljeholm et al., 2013*; *Sgrò et al., 2016*; *Taniguchi et al., 2014*; *Muzzi et al., 2009*; *Giansanti et al., 2004*; *Paw et al., 2003*; *Menon et al., 2014*; *Morita et al., 2005*; *Jackson et al., 2011*; *Di Cunto et al., 2000*; *LoTurco et al., 2003*; *Ackman et al., 2007*). In human patients, genome-wide association studies have revealed that genomic mutations in cytokinesis genes lead to cell-type-specific division failure and cell- or tissue-type-specific pathologies. For example, an autosomal dominant mutation in the human kinesin-6 MKLP1, a protein thought to be essential for cytokinesis in all animal cells (*Glotzer, 2009*), leads to congenital dyserythropoietic anemia type III (*Liljeholm et al., 2013*). These patients have multinucleated erythroblasts due to a failure in cytokinesis but are otherwise asymptomatic, indicating that cells in other tissues and organs divide successfully (*Liljeholm et al., 2013*). Mutations in Citron Kinase are associated with cytokinesis failure specifically in neuronal precursor cells, leading to multinucleated neurons and microcephaly in mice, rats, and human patients, but the development of other tissues and organs is not grossly disrupted (*Sgrò et al., 2016*; *Di Cunto et al., 2000*; *Ackman et al., 2007*; *Harding et al., 2016*; *Li et al., 2016*; *Basit et al., 2016*; *Shaheen et al., 2016*). Moreover, a mouse knockout of the microtubule

and actin-binding protein GAS2L3 dies shortly after birth due to specific defects in cardiomyocyte cytokinesis during heart development, but the overall development of other tissues is not affected (*Stopp et al., 2017*). These findings suggest that cell-type-specific mechanisms modulate the diversity of cytokinesis in animal cells, but the cellular and molecular mechanisms that underlie this diversity are poorly understood, even at a basic level.

There are two fundamental cell-type-specific regulatory mechanisms that could underlie cytokinetic variation: cell-intrinsic regulation (e.g. cell polarity, inherited proteins and RNAs) and cell-extrinsic regulation (e.g. cell-fate signaling, cell-cell adhesions). There is some evidence in support of each model. In support of cell-intrinsic regulation, we and others found that cell polarity proteins can promote robust cytokinesis during asymmetric cell division (*Jordan et al., 2016*; *Cabernard et al., 2010*; *Roth et al., 2015*), and asymmetrically inherited RNA granules in germ lineage cells contain a key splicing regulator involved in cytokinesis (*Audhya et al., 2005*). In support of extrinsic regulation, cell-cell adhesion junctions have been implicated in regulating contractile ring constriction in epithelial cells (*Guillot and Lecuit, 2013*; *Founounou et al., 2013*; *Herszterg et al., 2013*; *Bourdages and Maddox, 2013*; *Pinheiro et al., 2017*; *Lázaro-Diéguez and Müsch, 2017*; *Wang et al., 2018*; *Daniel et al., 2018*), and cell-fate signaling molecules, such as Wnt and Src, have been linked to cytokinesis in some contexts (*Fumoto et al., 2012*; *Kasahara et al., 2007*; *Soeda et al., 2013*). Thus, both cell-intrinsic and -extrinsic regulatory mechanisms could contribute to cel- type-specific diversity in cytokinesis.

The *C. elegans* four-cell embryo is a powerful, optically clear system to probe the mechanisms of cell-type-specific variation in cytokinesis. All four-cell divisions occur within a ~20 min time frame, and cytokinesis can easily be monitored in each individual blastomere, or cell within the embryo, by light microscopy. Worm development follows a defined cell lineage pattern (*Sulston and Horvitz, 1977*; *Sulston et al., 1983*), and the cell-fate patterning of each blastomere in the four-cell embryo is known (*Rose and Gönczy, 2014*). At the four-to-eight-cell division, each of the four cells are already specified to form distinct cell linages by conserved, well-characterized cell-fate signaling pathways (e.g. Notch/Delta, Wnt, Src; for review see [*Rose and Gönczy, 2014*; *Priess, 2005*; *Bowerman, 1995*]). The two-cell embryo divides to form two anterior blastomeres, ABa and ABp, and two posterior blastomeres, EMS and P2. While the anterior blastomeres are born as identical sisters, activation of Notch family receptors in ABp by a Delta-like ligand on the surface of P2 induces ABp to adopt a different cell fate than ABa (*Mickey et al., 1996*; *Bowerman et al., 1992*; *Mango et al., 1994*; *Moskowitz et al., 1994*; *Shelton and Bowerman, 1996*). The two posterior blastomeres, EMS and P2, are born from an asymmetric cell division and cell-cell contact-mediated Wnt and Src signaling between P2 and EMS promote asymmetric cell division and cell fate specification in both cells (*Goldstein, 1992*, *1993*, *1995a*, *1995b*; *Rocheleau et al., 1997*; *Thorpe et al., 1997*; *Bei et al., 2002*; *Arata et al., 2010*; *Schierenberg, 1987*). Therefore, in the four-cell *C. elegans* embryo, each cell has a unique cell identity and can be individually scored for contractile ring constriction during cytokinesis.

To study the mechanisms of cytokinetic variation, we combined thermogenetics with cell type-specific in vivo and ex vivo analysis of cytokinesis in each blastomere of the four-cell *C. elegans* embryo. We used fast-acting temperature-sensitive (ts) alleles to inactivate two cytokinesis proteins essential for contractile ring constriction in the one-cell embryo (*Davies et al., 2014*; *Liu et al., 2010*): the motor myosin-II (NMY-2 in *C. elegans*, hereafter myosin-II[NMY-2]) and the diaphanous formin CYK-1 (hereafter formin[CYK-1]). We identified cell-type-specific variation in four-cell embryos in the molecular requirement for formin[CYK-1], but not myosin-II[NMY-2]. Specifically, we found that while cytokinesis in two blastomeres, ABa and ABp, is sensitive to reduced formin[CYK-1] activity, cytokinesis in the other two blastomeres, EMS and P2, is resistant to defects in formin[CYK-1]-mediated F-actin assembly. Likewise, cytokinesis in ABa and ABp cells is sensitive to treatment with low doses of LatrunculinA (LatA), a drug that block F-actin polymerization, whereas cytokinesis in EMS and P2 cells is resistant to LatA. We further found that EMS and P2 cells have greatly reduced F-actin levels at the division plane upon formin[CYK-1] disruption, despite considerable and often successful equatorial constriction during cytokinesis, suggesting that cytokinesis in these cells is less dependent on F-actin in the contractile ring. To determine if EMS- and P2-specific variation in cytokinesis regulation is due to cell-intrinsic or -extrinsic regulation, we isolated individual blastomeres by embryo micro-dissection and examined the effect on cytokinesis when kept in isolation or when sister blastomeres were paired. We found that P2 cells are protected against cytokinesis failure after formin[CYK-1]

disruption even when isolated from the embryo, indicating cell-intrinsic regulation of cytokinesis, whereas EMS cells are not protected against cytokinesis failure upon isolation. EMS cytokinetic protection is restored upon pairing with P2, but not with ABa/ABp cells, indicating that cytokinesis in EMS is subject to cell-extrinsic regulation by P2. Finally, we found that cytokinesis in EMS is dependent on the proto-oncogenic tyrosine kinase Src[SRC-1], a critical player in EMS cell fate specification that is known to be activated in EMS by direct contact with P2 (*Bei et al., 2002*; *Arata et al., 2010*). This work establishes the *C. elegans* four-cell embryo as a system to study cytokinetic variation and demonstrates that both cell-intrinsic and -extrinsic regulations contribute to cell-type-specific diversity in cytokinetic mechanisms.

## Results

We first sought to identify variation in the regulation of cytokinesis in individual blastomeres within the four-cell *C. elegans* embryo (*Figure 1A*). To do this, we took a thermogenetic approach and used fast-acting ts alleles to weaken the contractile ring, while monitoring differences in cytokinesis between ABa, ABp, EMS, and P2 at increasing temperatures by spinning disc confocal microscopy. We used ts alleles of two contractile ring proteins known to be essential for cell division in most animal cell types (*Davies et al., 2014*; *Liu et al., 2010*; *Severson et al., 2002*; *Castrillon and Wasserman, 1994*; *Afshar et al., 2000*; *Bohnert et al., 2013*; *Moseley and Goode, 2005*; *Chang et al., 1997*; *Kiehart et al., 1982*): the motor myosin-II[NMY-2] (*nmy-2(ne3409ts)*, hereafter *myosin-II[nmy-2](ts)*) and the diaphanous formin[CYK-1] (*cyk-1(or596ts)*, hereafter *formin[cyk-1](ts)*). The *myosin-II[nmy-2](ts)* mutant has a point mutation in the myosin neck (S2) domain, required for dimerization and head coupling (*Liu et al., 2010*; *Tama et al., 2005*). The *formin[cyk-1](ts)* mutant has a point mutation in the post-region (FH2 domain) required for dimerization and processive F-actin polymerization (*Pruyne et al., 2002*), including at the division plane (*Figure 1—figure supplement 1*) (*Davies et al., 2014*). Both mutations completely block cytokinesis in the *C. elegans* one-cell embryo and have a null-like phenotype at restrictive temperature (26°C), with no contractile ring constriction (*Davies et al., 2014*).

These ts mutants are functionally tunable with temperature, having higher activity at lower temperatures and lower activity at higher temperatures (*Davies et al., 2017*). We used this property of thermal tunability to determine if the requirement for myosin-II[NMY-2] or formin[CYK-1] varies between the four different blastomeres. We upshifted ts mutant four-cell embryos from a permissive temperature (16°C) to a higher temperature across a thermal range up to restrictive temperature (18–26°C) before anaphase onset and monitored the cytokinetic phenotype at that temperature (*Figure 1B*, *Video 1* and see Materials and methods). In control embryos, all four blastomeres successfully completed cytokinesis after pre-anaphase upshift across the range of temperatures tested (*Figure 1B,C*). In *myosin-II[nmy-2](ts)* embryos, individual blastomeres within the four-cell embryo exhibited a similar frequency of cytokinesis failure after pre-anaphase upshift to higher temperatures across the range of upshift temperatures tested (*Figure 1B,C*), although ABa and ABp failed in cytokinesis at a slightly lower frequency than EMS and P2 at intermediate temperatures (20–22°C; *Figure 1C*). Thus, we found that decreased levels of myosin-II[NMY-2] activity caused a similar frequency of cytokinesis failure in all four blastomeres.

In contrast, *formin[cyk-1](ts)* embryos showed substantial blastomere-specific differences in cytokinesis failure after upshift to increasing temperatures. ABa and ABp cells from *formin[cyk-1](ts)* embryos were similar to the one-cell embryo (*Davies et al., 2014*): they started to fail in cytokinesis at 19 and 21°C, respectively, and both blastomeres failed in cytokinesis 100% of the time when upshifted to fully restrictive temperature (26°C; *Figure 1C*). In contrast, EMS and P2 cells from *formin[cyk-1](ts)* embryos were relatively resistant to cytokinesis failure. Both blastomeres successfully completed cytokinesis 100% of the time below ~24°C and with high frequency even at 26°C, a temperature at which ABa and ABp always fail in cytokinesis (5/9 EMS and 4/11 P2 cells, versus 0/5 ABa and 0/5 ABp cells, complete cytokinesis at 26°C; *Figure 1C*). In fact, even at 27°C, a temperature at which even control worms start to show developmental defects due to thermal stress, ~40% of pre-anaphase upshifted EMS and P2 cells were still able to divide in *formin[cyk-1](ts)* embryos (5/12 EMS and 7/17 P2 cells, versus 0/10 ABa and 0/10 ABp cells, complete cytokinesis at 27°C; *Figure 1—figure supplement 2*). Together, these data show that cytokinesis in EMS and P2 requires lower levels of formin[CYK-1] activity than cytokinesis in ABa and ABp cells (or the one-cell embryo, see [*Davies et al.,*

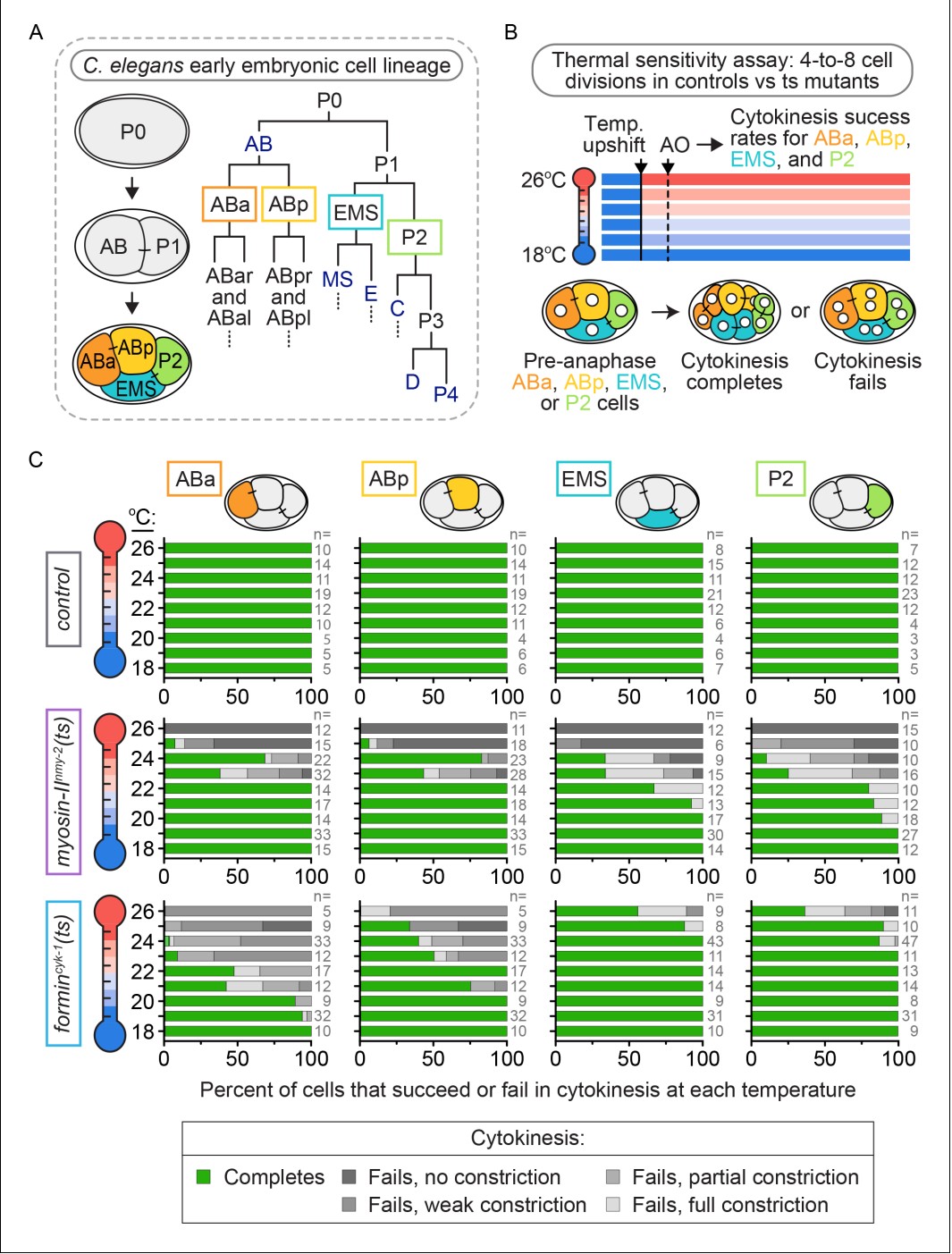

**Figure 1.** Cytokinetic variation with loss of formin[CYK-1], but not myosin-II[NMY-2], activity in individual blastomeres of the four-cell embryo. (**A**) Lineage map showing the identity and division patterning that occurs during the early blastomere divisions in the *C. elegans* embryo. Founder cells AB, E, MS, C, D, and P4 are indicated in dark blue. (**B**) Schematic of the experimental protocol for the thermal sensitivity assay. Individual ts mutant (or control) four-cell embryos were upshifted from permissive temperature (16°C) to a higher temperature across a thermal range (18–26°C) prior to anaphase onset in each blastomere. (**C**) Graphs showing the cytokinetic outcome for control, *myosin-II[nmy-2](ts)*, and *formin[cyk-1](ts)* mutant embryos upshifted to specific temperatures prior to anaphase onset. AO = anaphase onset. The percent of cells exhibiting each cytokinetic phenotype at the indicated temperature is plotted for each cell type and genotype. n ≥ 81 for each cell type (detailed in *Supplementary file 1*).

DOI: https://doi.org/10.7554/eLife.36204.003

*Figure 1 continued on next page*

*Figure 1 continued*

The following figure supplements are available for figure 1:

**Figure supplement 1.** F-actin contractile ring assembly is disrupted in *formin$^{cyk-1}$(ts)* mutant embryos at restrictive temperature.

DOI: https://doi.org/10.7554/eLife.36204.004

**Figure supplement 2.** EMS and P2 can divide in the absence of formin$^{CYK-1}$ activity.

DOI: https://doi.org/10.7554/eLife.36204.005

*2014*]), and suggests that cytokinesis in these cells is differentially regulated at a cell-type-specific level.

To test for differences in the temporal requirements for these contractile ring proteins between the individual blastomeres, we next performed temperature upshifts from permissive (16°C) to fully restrictive (26°C) temperature with *myosin-II$^{nmy-2}$(ts)* and *formin$^{cyk-1}$(ts)* mutant four-cell embryos at specific times before and after anaphase onset and monitored the frequency of cytokinesis failure in ABa, ABp, EMS, and P2 cells (*Figure 2A*, see Materials and methods). In control embryos, all four blastomeres successfully completed cytokinesis 100% of the time, irrespective of when the thermal upshifts occurred (n ≥ 57 for each cell type; *Figure 2B*). In *myosin-II$^{nmy-2}$(ts)* embryos, all four blastomeres were similar to the one-cell embryo and failed in cytokinesis 100% of the time whether upshifted before anaphase onset or later, up to ≤10 min after anaphase onset (just prior to contractile ring closure) (n ≥ 67 for each cell type; *Figure 2B*) (*Davies et al., 2014*). Thus, we found that myosin-II$^{NMY-2}$ activity is temporally required throughout cytokinesis in all four blastomeres.

In contrast, *formin$^{cyk-1}$(ts)* embryos again showed cell type variation in the temporal requirement for activity among individual blastomeres. In our previous analysis of the one-cell *C. elegans* embryo, cytokinesis failed 100% of the time with formin$^{CYK-1}$ inactivation by thermal upshift to restrictive temperature at any time before mid-ring constriction (*Davies et al., 2014*). Consistent with this, ABa and ABp cells in *formin$^{cyk-1}$(ts)* embryos failed in cytokinesis 100% of the time when upshifted to 26°C prior to anaphase onset (n ≥ 61 for each cell type; *Figure 2B*). However, ABa and ABp cells differed in their temporal requirement for formin$^{CYK-1}$ activity when upshifted ≤10 min after anaphase onset: while ABa cells failed in cytokinesis 100% of the time, ABp cells failed only ~50% of the time (n ≥ 61 for each cell type; *Figure 2B*). EMS and P2 cells in *formin$^{cyk-1}$(ts)* mutant embryos exhibited even more dramatic differences in cytokinesis failure. A number of these cells successfully divided when upshifted to 26°C well before anaphase onset (14/43 EMS and 16/50 P2 cells complete cytokinesis) (*Figures 2B and C*) and showed substantial equatorial constriction and frequent successful cytokinesis when upshifted to 26°C after anaphase onset (*Figure 2B*; *Figure 2—figure supplement 1*). Thus, while formin$^{CYK-1}$ activity is essential in ABa and ABp, with ABa requiring high functional levels of formin$^{CYK-1}$ activity throughout the entirety of cytokinesis and ABp requiring formin$^{CYK-1}$ activity only early in contractile ring assembly and constriction (like in the 1 cell embryo, see [*Davies et al., 2014*]), EMS and P2 cells have lower overall requirements for formin$^{CYK-1}$ activity. Taken together, our functional and temporal requirement analysis suggests that differences in formin$^{CYK-1}$-mediated actin dynamics may underlie cell-type-based cytokinetic diversity.

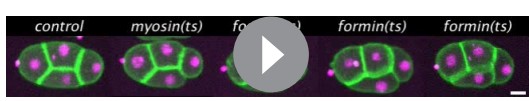

**Video 1.** Control, *myosin-II$^{nmy-2}$(ts)*, and *formin$^{cyk-1}$(ts)* mutant embryos undergoing cytokinesis at the restrictive temperature. The three *formin$^{cyk-1}$(ts)* mutant embryos show phenotypic variation in the cytokinesis outcome for EMS and P2, as is observed at this temperature. 60 s per frame; temperature, 26°C. Green, GFP::PH (plasma membrane); magenta, mCherry:: histone2B; scale bar = 10 μm.

DOI: https://doi.org/10.7554/eLife.36204.006

In animal cells, diaphanous family formin-mediated assembly of linear F-actin during cytokinesis is thought to be essential for the assembly and constriction of the actomyosin contractile ring (*D'Avino et al., 2015*; *Pollard, 2010*; *Severson et al., 2002*; *Bohnert et al., 2013*). Formin$^{CYK-1}$ is the only worm diaphanous family formin (*Pruyne, 2016*). In *formin$^{cyk-1}$(ts)* one-cell *C. elegans* embryos at 16°C, F-actin is present in the contractile ring (although at lower levels than in control embryos) but, upon upshift to 26°C, linear F-actin is no longer visible and cytokinesis fails (*Figure 1—figure supplement 1* and [*Davies et al., 2014*]). It is possible that in

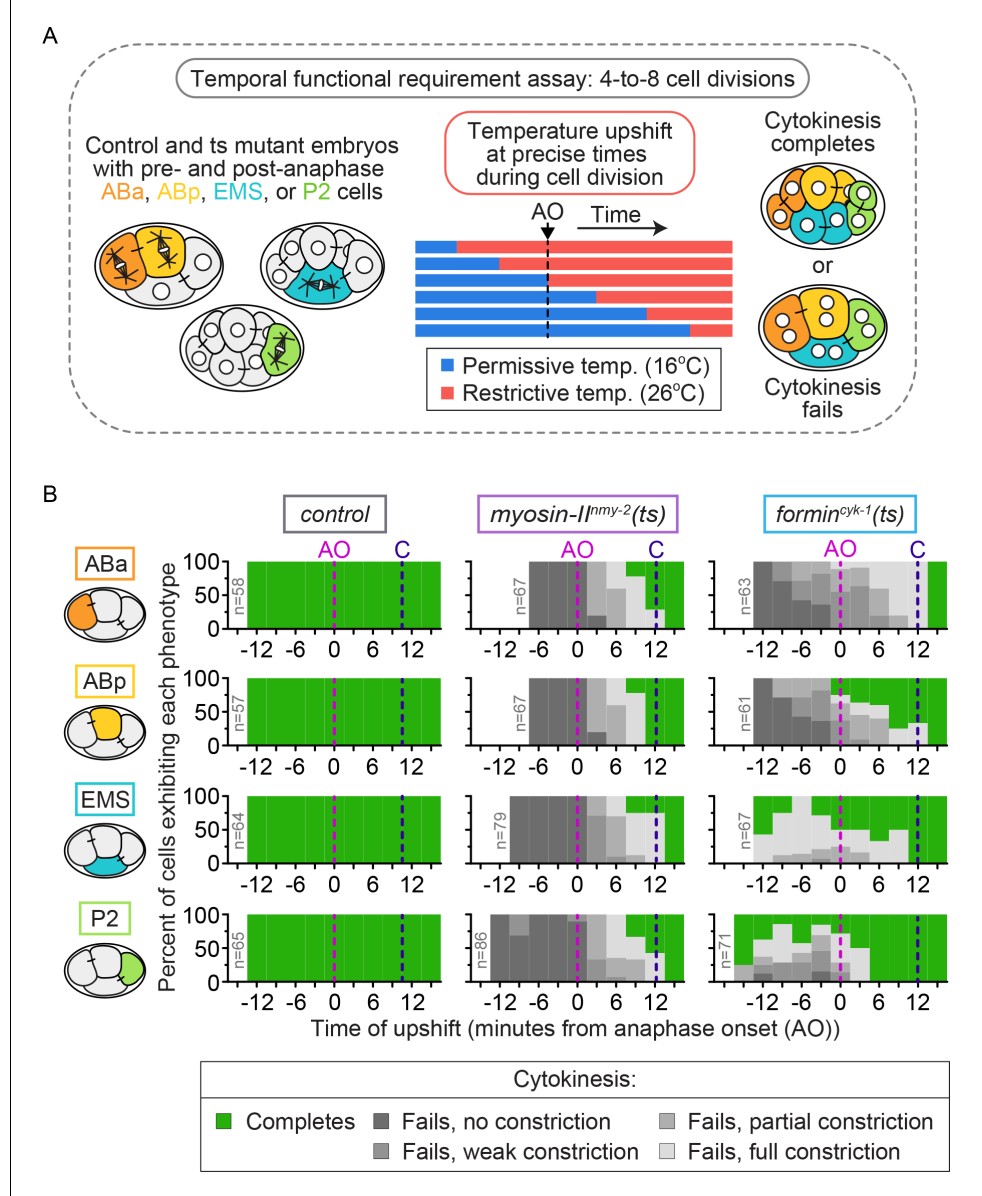

**Figure 2.** Cytokinetic variation in the temporal requirement for formin[CYK-1], but not myosin II[NMY-2], in individual blastomeres of the four-cell embryo. (**A**) Schematic of the experimental protocol for the temporal functional requirement assay in which four-cell stage embryos were upshifted from permissive (16°C) to restrictive temperature (26°C) at defined time points relative to anaphase onset in each individual blastomere, then held at 26°C throughout cytokinesis. (**B**) Graphs showing the cytokinetic outcome for control, *myosin-II[nmy-2](ts)*, and *formin[cyk-1](ts)* mutant embryos upshifted from 16°C to 26°C at different times during cell division. The cytokinetic outcome of each cell is plotted as a percent of the total number of cells upshifted to 26°C at that time relative to anaphase onset. AO = anaphase onset (magenta dashed line); C = approximate time of contractile ring closure at 16°C (dark blue dashed line); n ≥ 57 for each cell type (see *Supplementary file 1*).
DOI: https://doi.org/10.7554/eLife.36204.007

The following figure supplement is available for figure 2:

**Figure supplement 1.** Loss of formin[CYK-1] slows division in EMS and P2 cells.
DOI: https://doi.org/10.7554/eLife.36204.008

EMS and/or P2, another formin-related protein could function redundantly with formin[CYK-1] to ensure F-actin assembly during cytokinesis in these specific cells. Transcriptional analysis has revealed there are two other formin-related genes (*inft-2* and *frl-1)* expressed at the four-cell stage

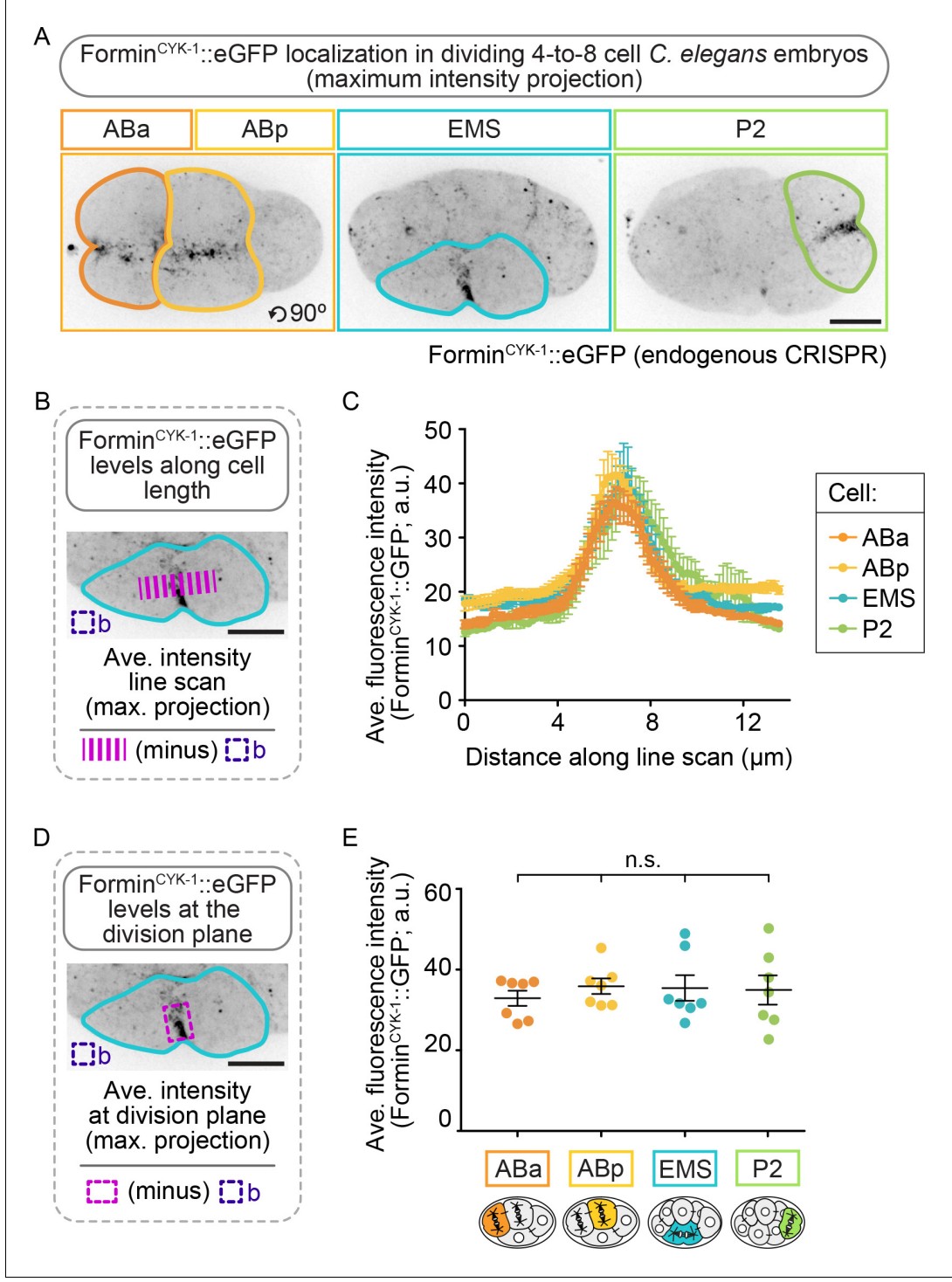

**Figure 3.** Formin[CYK-1] localizes to the contractile ring at similar peak levels in the ABa, ABp, EMS and P2 cells. (**A**) Representative maximum intensity projection images showing formin[CYK-1]::GFP localization at the division plane in ABa and ABp (left panel), EMS (center panel), and P2 (right panel). Images were acquired after observation of the onset of contractile ring constriction (initial furrowing). (**B**) Schematic showing how formin[CYK-1]::GFP levels were measured along a line scan across the division plane. (**C**) Graph showing all four cells show a local peak in the level of formin[CYK-1]::GFP at the division plane. (**D**) Schematic showing how formin[CYK-1]::GFP levels at the division plane were measured. (**E**) Graph showing the average fluorescence maximum intensity of formin[CYK-1]::GFP at the contractile ring is not significantly different between ABa, ABp, EMS, or P2. Two-tailed t-test (**Supplementary file 1**); n.s., no significance, $p > 0.05$. Error bars, mean ± SEM; temperature, 21°C; scale bar = 10 μm.

*Figure 3 continued on next page*

*Figure 3 continued*

DOI: https://doi.org/10.7554/eLife.36204.009

The following figure supplements are available for figure 3:

**Figure supplement 1.** Depletion of other formin-related genes does not prevent cytokinesis in EMS and P2 cells following formin^CYK-1 inhibition.

DOI: https://doi.org/10.7554/eLife.36204.010

**Figure supplement 2.** Generation of tagged formin^CYK-1::eGFP at the endogenous locus.

DOI: https://doi.org/10.7554/eLife.36204.011

**Figure supplement 3.** Sum projection analysis of formin^CYK-1::GFP localization during cytokinesis in the ABa, ABp, EMS and P2 cells.

DOI: https://doi.org/10.7554/eLife.36204.012

(*Hashimshony et al., 2015*), but no formin-related gene is significantly over-expressed in either EMS or P2, relative to in ABa or ABp (*Tintori et al., 2016*). To test if other formin-related proteins could compensate for loss of formin^CYK-1 activity during cytokinesis in EMS and P2, we depleted the six other *C. elegans* formin-related proteins by RNAi in *formin^cyk-1(ts)* mutants and monitored the success or failure of cytokinesis in the four-cell embryo. If another formin-related protein compensates for a loss of formin^CYK-1 activity, then reducing the levels of that formin should increase the frequency of cytokinesis failure in EMS and P2. Instead, we found that individual depletion of the other formin-related proteins did not decrease the frequency cytokinesis failure in any of the four blastomeres in *formin^cyk-1(ts)* mutants (*Figure 3—figure supplement 1*). This suggests that no single formin-related protein is compensating for loss of formin^CYK-1 activity, although we cannot rule out that multiple formin-related proteins may function together during cytokinesis in EMS and P2 specifically.

Although *formin^cyk-1* mRNA levels do not vary across the four blastomeres (*Tintori et al., 2016*), it is possible that formin^CYK-1 protein levels are higher in EMS and/or P2, thus leading to higher formin^CYK-1 activity and higher resistance to partial inactivation of formin^CYK-1 function in these cells. To test this possibility, we tagged the C-terminus of formin^CYK-1 at the endogenous locus using a CRISPR/Cas9 method (*Dickinson et al., 2015*) and generated a homozygous *C. elegans* strain expressing formin^CYK-1::eGFP (*Figure 3—figure supplement 2*). Using this strain, we imaged the four cell types after formation and partial ingression of the contractile ring in each cell type and found that formin^CYK-1::eGFP localized nearly exclusively to the division plane during cytokinesis in all four blastomeres. Analysis of the maximum intensity projection (which captures the cortical eGFP signal) revealed that formin^CYK-1::eGFP is present at similar peak levels at the division plane in all four cells (*Figure 3*). Sum intensity projection analysis (total levels) revealed that formin^CYK-1::eGFP is reduced in EMS and P2, versus ABa/ABp blastomeres (*Figure 3—figure supplement 3*). While the challenges of sum intensity projection analysis in multicellular embryos make it difficult to form conclusions about protein levels using this approach (see Materials and methods for additional information), these results are consistent with the maximum intensity projection analysis and do not support the hypothesis that resistance to formin^CYK-1 inactivation in EMS and P2 is due to an endogenous enrichment of formin^CYK-1 protein in these cells relative to ABa and ABp at the four-cell stage.

If multiple formin-related proteins function together to promote contractile ring assembly in EMS and P2 when formin^CYK-1 activity is reduced, then F-actin at the division plane should remain at high levels in these cells upon temperature upshift in *formin^cyk-1(ts)* mutant embryos. This outcome would also be predicted if stable, pre-existing F-actin filaments, rather than de novo formin^CYK-1-nucleated filaments, comprise the contractile ring during cytokinesis in EMS and P2. Thus, we next measured contractile ring F-actin levels in *formin^cyk-1(ts)* mutant embryos after the onset of equatorial constriction at the four-cell stage using the F-actin reporters Lifeact::RFP, PLST-1::GFP, and GFP::Utrophin^ABD (*Jordan et al., 2016*; *Riedl et al., 2008*; *Burkel et al., 2007*; *Ding et al., 2017*; *Tse et al., 2012*) to label the entire F-actin cytoskeleton including the contractile ring (*Figure 4*; *Figure 4—figure supplements 1–3*). F-actin was enriched at cell-cell junctions and at the cell division plane in all four blastomeres in control embryos at both 16°C and 26°C and in *formin^cyk-1(ts)* embryos at permissive temperature, 16°C (*Figure 4*; *Figure 4—figure supplements 1–3*). In *formin^cyk-1(ts)* embryos at restrictive temperature (26°C), F-actin was still present at the cell-cell junctions but was not enriched at the contractile ring during cell division in any of the four blastomeres, including in EMS and P2 cells that successfully completed cytokinesis (*Figure 4*; *Figure 4—figure supplements 1–3*). It is

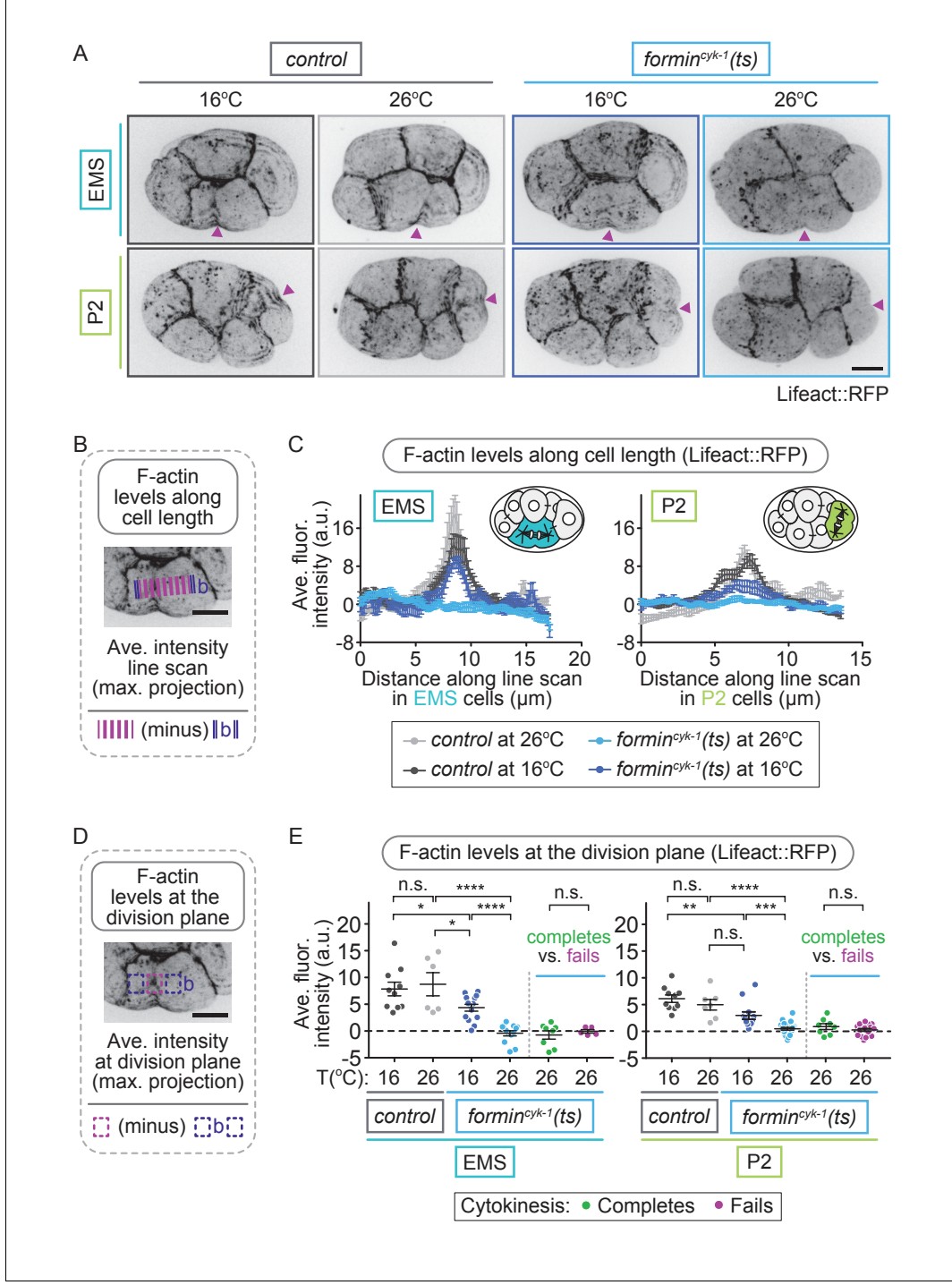

**Figure 4.** EMS and P2 cells can divide in the absence of a robust F-actin contractile ring. (**A**) Representative images showing Lifeact::RFP-labeled contractile ring F-actin can be seen in EMS and P2 cells in control embryos at 16 and 26°C and in formin[cyk-1](ts) embryos at 16°C, but not in formin[cyk-1](ts) embryos at 26°C. Arrowheads (magenta) indicate the division plane/site of initial furrowing. (**B**) Schematic showing how F-actin levels were measured along a line scan across the division plane in EMS and P2 cells. Images were acquired after observation of the onset of contractile ring constriction (initial furrowing). (**C**) Graphs showing line scans across EMS and P2 cells have a local peak in Lifeact::RFP-labeled F-actin at the division plane in control embryos at 16 and 26°C and formin[cyk-1](ts) embryos at 16°C, but not in formin[cyk-1](ts) embryos at 26°C. (**D**) Schematic showing how F-actin levels at the division plane were measured in EMS and P2 cells. Images were acquired after observation of the onset of contractile ring constriction (initial furrowing). (**E**) Graphs showing the average fluorescence intensity of Lifeact::RFP

*Figure 4 continued on next page*

*Figure 4 continued*

at the division plane in EMS and P2 cells is significantly decreased in *formin^cyk-1(ts)* embryos at 26°C, compared to at 16°C, or compared to in control embryos at 16 or 26°C. There was no significant difference between the average maximum fluorescence intensity of Lifeact::RFP at the division plane in EMS and P2 cells in *formin^cyk-1(ts)* embryos at 26°C that successfully complete cytokinesis versus in those that fail to divide. Two-tailed t-test (**SupplementaryfFile 1**); n.s., no significance, p>0.05; *p≤0.05; **p≤0.01; ****p≤0.0001. Error bars, mean ± SEM, scale bar = 10 μm.
DOI: https://doi.org/10.7554/eLife.36204.013

The following figure supplements are available for figure 4:

**Figure supplement 1.** AB daughter cells fail to divide in the absence of a robust F-actin contractile ring.
DOI: https://doi.org/10.7554/eLife.36204.014

**Figure supplement 2.** EMS and P2 cells can divide in the absence of a robust F-actin contractile ring.
DOI: https://doi.org/10.7554/eLife.36204.015

**Figure supplement 3.** EMS and P2 cells can divide in the absence of a robust F-actin contractile ring.
DOI: https://doi.org/10.7554/eLife.36204.016

---

important to note that these *C. elegans* F-actin reporters are dim, even in control embryos; thus, our inability to detect F-actin on our microscope system with these reporters does not indicate an absence of F-actin in the contractile ring. Nonetheless, together with our formin-related protein RNAi mini-screen results (**Figure 3—figure supplement 1**), the decrease in robust F-actin levels in all cells in *formin^cyk-1(ts)* embryos at restrictive temperature suggests that equatorial constriction and successful cytokinesis in EMS and P2 is not likely due to F-actin assembly by another formin-related protein or due to utilization of pre-existing stabilized F-actin to form the contractile ring. Instead, it suggests cell-type-specific mechanism(s) allow cytokinesis to occur in the absence of a robust F-actin cytoskeleton in these blastomeres.

To determine if robust cytokinesis in EMS and P2 could also withstand perturbations of the F-actin cytoskeleton independent of formin^CYK-1 activity, we next tested whether cytokinesis in EMS and P2 is resistant to pharmacological inhibition of F-actin polymerization by treatment with low doses of the F-actin inhibitor, LatrunculinA (LatA). Embryos were first permeabilized by *perm-1 (RNAi)*, a gene required for normal eggshell assembly (**Carvalho et al., 2011**; **Olson et al., 2012**). Control, non-ts mutant, four-cell stage embryos were incubated with growth medium containing different concentrations of LatA and the lipophilic dye FM 4-64 for at least 10 min before anaphase onset and observed undergoing cytokinesis. Only embryos showing FM 4-64 on the plasma membrane (indicating eggshell permeability), were included in the analysis (**Figure 5A**). In embryos treated vehicle control (DMSO only, 0 nM LatA), all four blastomeres divided 100% of the time (**Figure 5B**). However, in embryos treated with increasing concentrations of LatA (50, 67, and 80 nM), we observed cell-specific effects of F-actin inhibition, with EMS and P2 cells always dividing at higher frequencies than ABa and ABp cells treated with the same LatA concentration (**Figure 5B**). For example, in embryos treated with 80 nM LatA, 100% of ABa (n = 13) and ABp (n = 13) cells failed in cytokinesis, while EMS and P2 cells divided ~25% of the time (EMS: 26%, n = 19; P2: 28%, n = 18). These LatA results phenocopy the cell-type-specific requirement for formin^CYK-1 activity and again suggest that EMS and P2 are protected against cytokinesis failure when the F-actin cytoskeleton is weakened.

We next investigated whether protection from cytokinesis failure in EMS and P2 is due to cell-intrinsic or -extrinsic regulatory mechanisms. We first eliminated the potential for cell-extrinsic regulation by isolating each individual blastomere from embryos by manual microdissection (**Figure 6A**). After removing the eggshell, individual blastomeres were separated at the two-cell stage and allowed to divide again at permissive temperature, followed by another round of sister cell separation, resulting in an isolated ABa/ABp, EMS, and P2 cell from each embryo (**Figure 6A**). Upon isolation, the ABa and ABp cells cannot be distinguished from each other, as they are identical in size and fate in the absence of cell-contact-mediated extrinsic signaling from P2 to ABp (**Mickey et al., 1996**; **Bowerman et al., 1992**; **Mango et al., 1994**; **Moskowitz et al., 1994**; **Shelton and Bowerman, 1996**); hence, we refer to these isolated blastomeres as AB daughters (ABd) (**Figure 6A,B**). EMS and P2 can easily be distinguished from each other and the ABd by their unique sizes, which do not change upon isolation (**Figure 6—figure supplement 1**). When upshifted to 26°C, all isolated

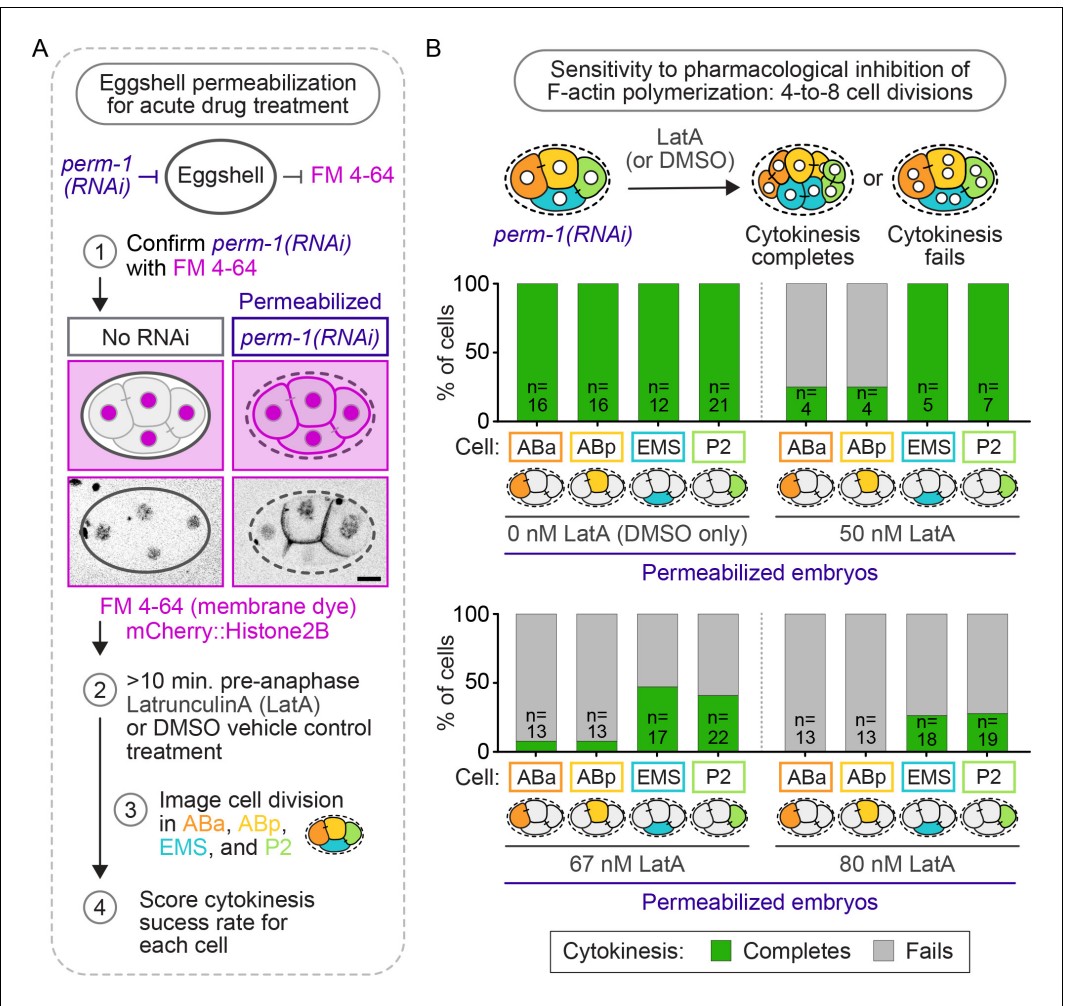

**Figure 5.** Cytokinesis in EMS and P2 is more resistant to pharmacological inhibition of F-actin assembly with LatA than in ABa and ABp. (**A**) Schematic of the experimental protocol for eggshell permeabilization with *perm-1(RNAi)*, confirmation of eggshell permeabilization with FM 4–64, and subsequent Latrunculin A (LatA) treatment of permeabilized control, non-ts, embryos. Scale bar = 10 μm. (**B**) Schematic of the experimental protocol and graphs showing the cytokinetic outcome for each cell in permeabilized four-cell embryos treated with 0, 50, 67, and 80 nM LatA. Temperature, 21°C. See also ***Supplementary file 1***.
DOI: https://doi.org/10.7554/eLife.36204.017

blastomeres from control and *formin<sup>cyk-1</sup>(ts)* mutant embryos entered mitosis with characteristic cell-cycle synchrony (ABd cells always divided together) and timing (first the Abd cells divide, then EMS, then P2) (***Video 2***). Control blastomeres were able to successfully complete cytokinesis in all cases (10/10 ABd, 10/10 EMS, 12/12 P2 cells complete cytokinesis), while blastomeres isolated from *formin<sup>cyk-1</sup>(ts)* mutant embryos showed cell-type-specific variation in the frequency of cytokinesis failure (***Figure 6B,D***; ***Video 3***). Similar to equivalent cells in intact embryos (***Figure 6C***), isolated ABd blastomeres from *formin<sup>cyk-1</sup>(ts)* embryos always failed in cytokinesis (0/26 ABd cells complete cytokinesis) and isolated P2 cells divided ~30% of the time (7/22 P2 cells complete cytokinesis) (***Figure 6B, D***). However, in contrast to in the intact embryo (***Figure 6C***) in which EMS cells divide ~33% of the time, isolated EMS blastomeres from *formin<sup>cyk-1</sup>(ts)* mutants never successfully completed cytokinesis when upshifted to restrictive temperature prior to anaphase onset (0/22 isolated EMS cells complete cytokinesis) (***Figure 6B,D***). Thus, upon elimination of cell-extrinsic regulation by blastomere isolation, P2 cells still completed cytokinesis at a frequency similar to P2 and EMS cells in the intact embryo, whereas isolated EMS blastomeres always failed in cytokinesis. These results suggest that cell-type-

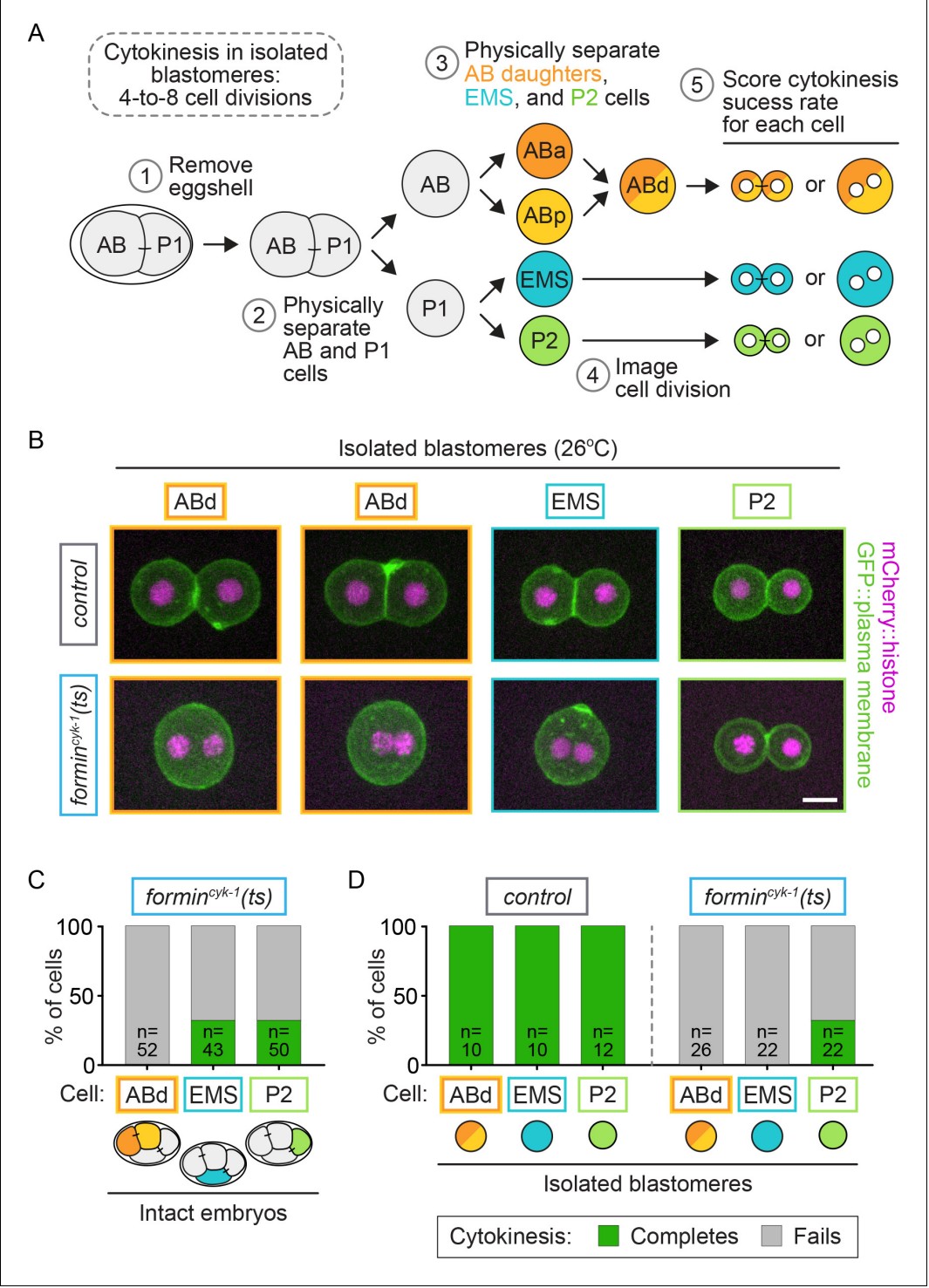

**Figure 6.** Cell-intrinsic and extrinsic regulation contribute to cytokinesis. (**A**) Experimental protocol describing the microdissection, isolation, and separation of individual blastomeres. Steps 1–3 are performed at the permissive temperature (16°C) to ensure the first two-cell divisions occur normally. (**B**) Representative images showing the cytokinetic outcome of cells isolated from control and *formin*$^{cyk-1}$*(ts)* embryos. Cells that divide successfully are seen as two mononucleate daughter cells. Cells that fail in cytokinesis are seen as single binucleate cells. (**C**) Graph showing the frequency of successful cytokinesis in individual blastomeres in intact *formin*$^{cyk-1}$*(ts)* embryos upshifted prior to anaphase onset. Note: this data is sub-sampled from the temporally defined upshift experiments shown in **Figure 2B**, pooling only those cells upshifted before anaphase onset. ABa and ABp cells

*Figure 6 continued on next page*

*Figure 6 continued*
have been combined as AB daughters (ABd). (**D**) Graph showing the frequency of successful cytokinesis for blastomeres isolated from control and *formin$^{cyk-1}$(ts)* embryos. Temperature, 26°C; scale bar = 10 μm. See also *Supplementary file 1*.
DOI: https://doi.org/10.7554/eLife.36204.018
The following figure supplement is available for figure 6:

**Figure supplement 1.** Blastomere size in intact four-cell embryos and following isolation.
DOI: https://doi.org/10.7554/eLife.36204.019

specific variation in cytokinesis failure upon formin$^{CYK-1}$ disruption is controlled cell-intrinsically in P2 and cell-extrinsically in EMS.

During cell fate specification, EMS is cell-extrinsically controlled by cell-cell contact dependent signals from the neighboring P2 cell, which promote cell fate induction and spindle orientation in EMS (for review see [*Rose and Gönczy, 2014*]). To determine if direct contact between EMS and P2 regulates cytokinesis in *formin$^{cyk-1}$(ts)* EMS cells, we again isolated sister blastomeres at the two-cell stage but this time did not separate sister cells after the two-to-four cell divisions, leaving ABd-ABd and EMS-P2 paired-doublets intact (*Figure 7A*). After temperature upshift to restrictive temperature, all paired blastomeres from control embryos successfully completed cytokinesis, and all paired ABd-ABd blastomeres from *formin$^{cyk-1}$(ts)* mutant embryos failed to divide, as expected (*Figure 7B, C*; *Video 4*). In EMS-P2 paired blastomeres from *formin$^{cyk-1}$(ts)* mutants, 29% of P2 cells divided successfully (9/31 P2 cells complete cytokinesis), similar to isolated P2 blastomeres (*Figures 6D* and *7C*; *Video 4*). Furthermore, 16% of EMS blastomeres in paired EMS-P2 doublets divided successfully (5/31 EMS cells complete cytokinesis) (*Figure 7B,C*), in contrast to isolated EMS blastomeres (*Figure 6D*) or manually paired EMS-ABd blastomeres (0/15 EMS cells complete cytokinesis), which never divided successfully (*Figure 7C*). We did not see a correlation between the success of EMS and the success or failure of cytokinesis in P2, as sometimes both blastomeres successfully completed cytokinesis (*Figure 7B*), but other times both failed or one blastomere completed and the other failed in cytokinesis (e.g. *Video 5*). Thus, direct cell-cell contact from P2 is sufficient to mediate protection against cytokinesis failure in EMS.

While contact with P2 could rescue cytokinesis in EMS upon formin$^{CYK-1}$ disruption, the cytokinesis success frequency for EMS in paired EMS-P2 doublets was lower than in intact *formin$^{cyk-1}$(ts)* embryos (16 vs. 33% respectively; *Figures 7C* and *6C*). One possibility is that P2 to EMS cell-fate signaling is not as efficient in paired-blastomere doublets as it is in the intact embryo. EMS-P2 signaling is mediated by the interface between the two cells (*Goldstein, 1992*, *1993*, *1995a*, *1995b*; *Rocheleau et al., 1997*; *Thorpe et al., 1997*; *Bei et al., 2002*; *Arata et al., 2010*; *Schierenberg, 1987*; *Heppert et al., 2018*), and in intact embryos the cells are constrained by the eggshell, pushing them together and resulting in a large contact area between the cells that is reduced upon blastomere isolation. We hypothesized that an increased contact area may enhance EMS-P2 signaling and therefore calculated the cell-cell contact area of the EMS-P2 doublets (*Figure 7D*). In *formin$^{cyk-1}$(ts)* paired EMS-P2 doublets in which the EMS cell divided successfully, there was a significantly larger cell-cell contact area compared to in doublets in which the EMS cell failed in cytokinesis (p=0.0266; *Supplementary file 1*; *Figure 7E*). There was no significant difference in cell-cell contact area between paired doublets from control embryos and paired doublets

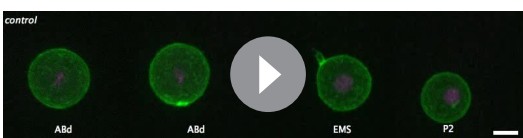

**Video 2.** Isolated blastomeres from a control embryo. 60 s per frame; temperature, 26°C. Green, GFP::plasma membrane; magenta, mCherry::histone2B; scale bar = 10 μm.
DOI: https://doi.org/10.7554/eLife.36204.020

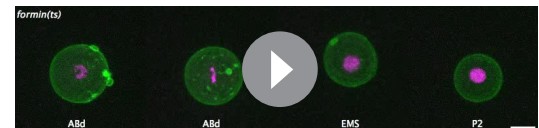

**Video 3.** Isolated blastomeres from a *formin$^{cyk-1}$(ts)* embryo. 60 s per frame; temperature, 26°C. Green, GFP::plasma membrane; magenta, mCherry::histone2B; scale bar = 10 μm.
DOI: https://doi.org/10.7554/eLife.36204.021

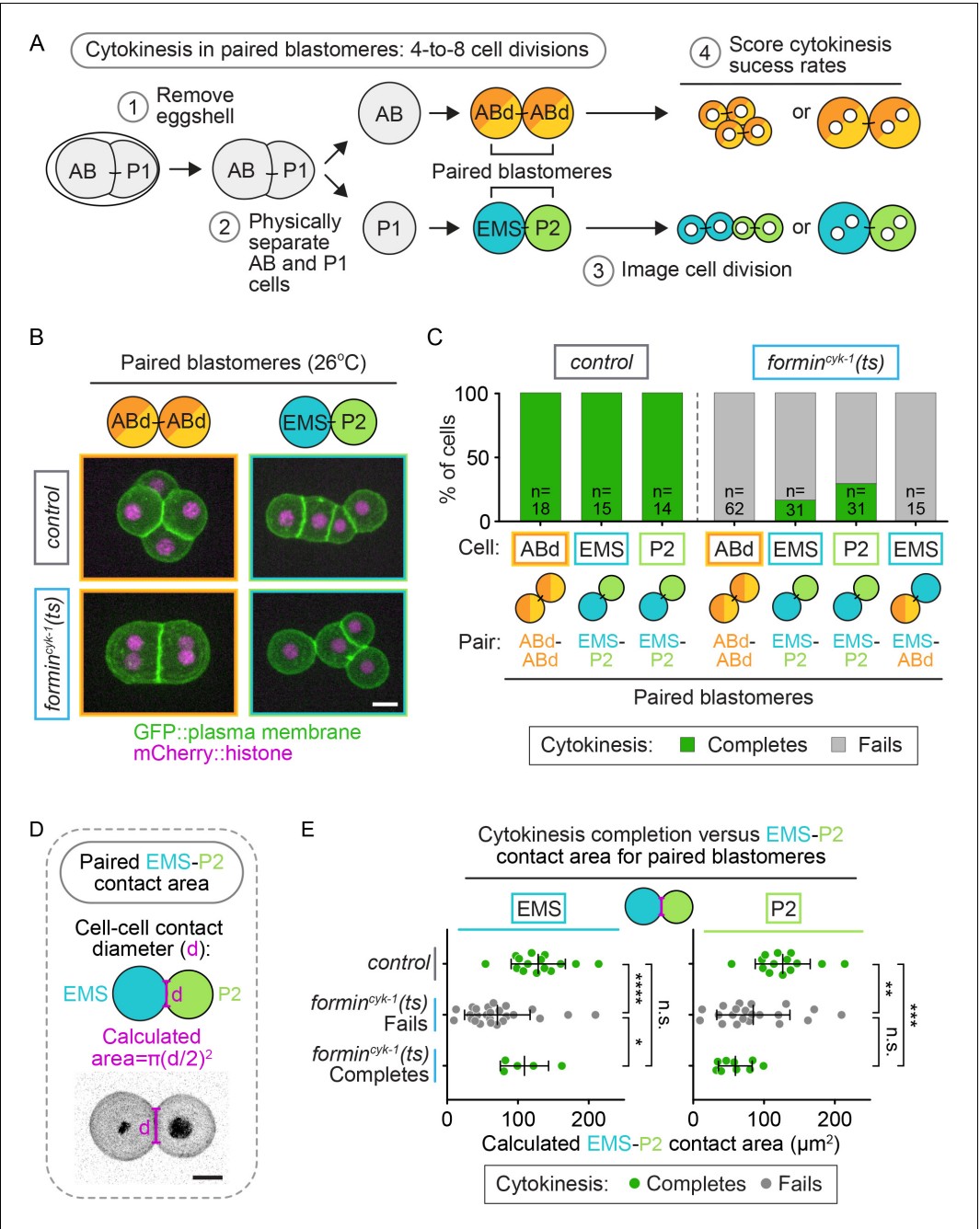

**Figure 7.** Cell-extrinsic regulation of cytokinesis in EMS depends on direct contact with its neighbor cell, P2. (**A**) Schematic of the experimental protocol for the isolation of paired-blastomere doublets. Blastomeres are maintained at the permissive temperature (16°C) during preparation of cell doublets (steps 1 and 2) to ensure the first two-cell divisions occur normally. (**B**) Representative images showing the cytokinetic outcome of paired blastomeres from control and *formin*<sup>cyk-1</sup>*(ts)* embryos. (**C**) Graph showing the frequency of successful cytokinesis for paired blastomeres isolated from control and *formin*<sup>cyk-1</sup>*(ts)* embryos. (**D**) Schematic showing the measurement of cell-cell contact diameter to calculate the cell-cell contact area between cells in EMS-P2 doublets (**E**). Graphs showing the calculated cell-contact area between EMS and P2 cells in paired doublets isolated from control and *formin*<sup>cyk-1</sup>*(ts)* embryos. Mean ± SD; two-tailed Mann-Whitney test (***Supplementary file 1***); n.s., no significance, p>0.05; *p≤0.05; **p≤0.01; ***p≤0.001; ****p≤0.0001. Temperature, 26°C; scale bar = 10 µm.
DOI: https://doi.org/10.7554/eLife.36204.022

from *formin*<sup>cyk-1</sup>*(ts)* mutants in which EMS divided successfully (p=0.3056; *Supplementary file 1*; *Figure 7E*). Because increased cell-cell contact area correlated with an increased probability of successful EMS division and because P2 to EMS signaling is dependent on cell-cell contact, this result suggests a correlation between successful P2 to EMS signaling and successful EMS division when formin<sup>CYK-1</sup> activity is reduced.

Another marker for successful P2-EMS signaling is spindle orientation in EMS cells. During extrinsic P2 to EMS cell-fate signaling, receptor tyrosine kinase (MES-1) and Wnt<sup>MOM-2</sup> from P2 activate Src<sup>SRC-1</sup> and Frizzled<sup>MOM-5</sup> proteins/receptors in EMS to promote differential cell fate specification and proper spindle orientation during the asymmetric EMS cell division (*Figure 8A*) (*Goldstein, 1992*, *1993*, *1995a*, *1995b*; *Rocheleau et al., 1997*; *Thorpe et al., 1997*; *Bei et al., 2002*; *Arata et al., 2010*; *Schierenberg, 1987*; *Heppert et al., 2018*). Therefore, we tested whether the success or failure of cytokinesis in EMS-P2 doublets from control and *formin*<sup>cyk-1</sup>*(ts)* mutants correlated with proper orientation of the EMS spindle angle, which reflects proper Src<sup>SRC-1</sup> and Frizzled-<sup>MOM-5</sup> signaling and successful cell fate specification (*Figure 8B*) (*Goldstein, 1992*, *1993*, *1995a*, *1995b*; *Rocheleau et al., 1997*; *Thorpe et al., 1997*; *Bei et al., 2002*; *Arata et al., 2010*; *Schierenberg, 1987*). In EMS-P2 doublets isolated from control embryos, which always divided successfully, the EMS spindle always aligned near the doublet axis (~180°), indicating successful P2 to EMS cell-fate signaling (*Figure 8C*). In *formin*<sup>cyk-1</sup>*(ts)* EMS-P2 doublets, EMS spindle orientation varied widely relative to the doublet axis with only 48% of *formin*<sup>cyk-1</sup>*(ts)* EMS spindles in alignment with the EMS-P2 doublet axis (>160° relative to the EMS-P2 doublet axis) (15/31 EMS cells; *Figure 8C*). This result suggests a potential role for formin<sup>CYK-1</sup> in this F-actin-dependent P2 to EMS signaling event (*Goldstein, 1995b*). In contrast, in *formin*<sup>cyk-1</sup>*(ts)* EMS cells that successfully completed cytokinesis, the EMS spindle axis was always aligned with the doublet axis (*Figure 8C*), consistent with successful P2 to EMS signaling (5/5 EMS cells) (*Bei et al., 2002*). Indeed, of the paired *formin*<sup>cyk-1</sup>*(ts)* EMS cells with normal spindle alignment, 33% (5/15 EMS cells; *Figure 8C*) divided successfully, a similar frequency to that observed in intact embryos. Consistent with the cell-intrinsic regulation of cytokinesis in P2, cytokinesis outcome in P2 was independent of EMS spindle orientation and thus P2-EMS cell-fate signaling (*Figure 8C*).

If signaling from the P2 cell contributes to EMS cell division, rather than just cell-cell contact, blocking the signal pathway in intact *formin*<sup>cyk-1</sup>*(ts)* embryos should decrease the frequency of successful EMS cell division. To test this, we used RNAi to deplete the non-receptor tyrosine kinase Src<sup>SRC-1</sup> in *formin*<sup>cyk-1</sup>*(ts)* mutant embryos and disrupt P2-mediated cell fate specification in EMS (*Bei et al., 2002*; *Arata et al., 2010*; *Sugioka and Sawa, 2010*) and then monitored cell division in each cell. *Src*<sup>src-1</sup>*(RNAi)* decreased the frequency of successful division in EMS (2/46 cells) compared with *control(RNAi)* (21/65 cells complete cytokinesis), while the frequency of successful division in P2 was unaffected (29/77 *control(RNAi)* P2 cells; 22/50 *Src*<sup>src-1</sup>*(RNAi)* P2 cells divided successfully) (*Figure 8D*). *Src*<sup>src-1</sup>*(RNAi)* had no effect on the frequency of successful division in non-ts control embryos (*Figure 8—figure supplement 1*). Thus, the extrinsic mechanism protecting EMS cells from cytokinesis failure after inhibition of *formin*<sup>cyk-1</sup> is dependent on Src<sup>SRC-1</sup> signaling

**Video 4.** Isolated ABd-ABd (left) and EMS-P2 (right) doublets from control (top) and *formin*<sup>cyk-1</sup>*(ts)* (bottom) embryos. 60 s per frame; temperature, 26°C. Green, GFP::plasma membrane; magenta, mCherry::histone2B; scale bar = 10 μm.
DOI: https://doi.org/10.7554/eLife.36204.023

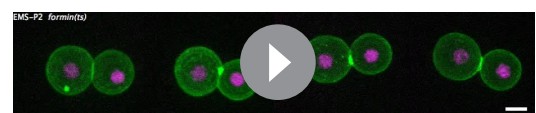

**Video 5.** Isolated EMS-P2 doublets from *formin*<sup>cyk-1</sup>*(ts)* embryos, showing combinations of cytokinesis phenotypes. 60 s per frame; temperature, 26°C; green, GFP::plasma membrane; magenta, mCherry::histone; scale bar = 10 μm.
DOI: https://doi.org/10.7554/eLife.36204.024

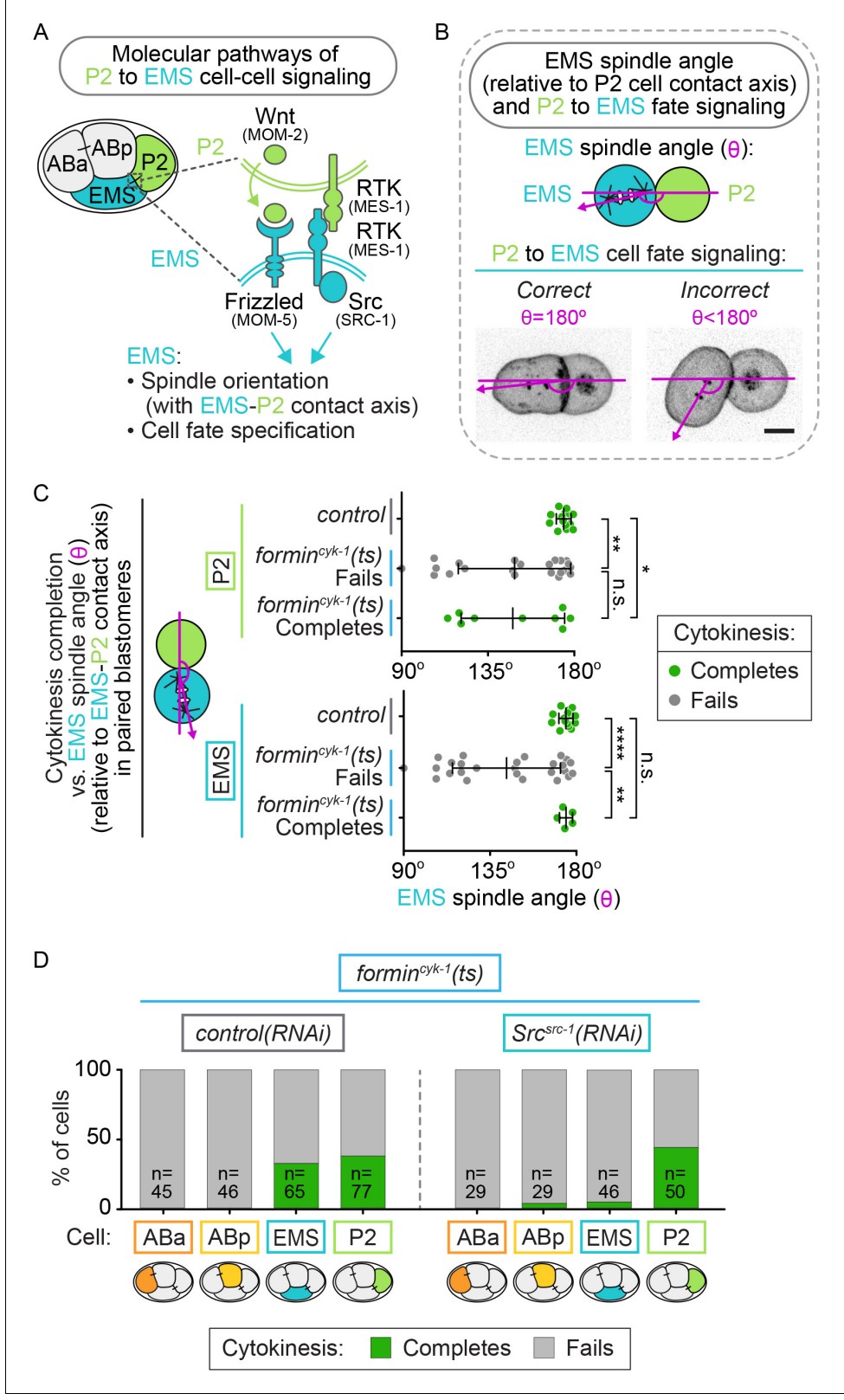

**Figure 8.** Src[SRC-1] mediated signaling from P2 provides the cell-extrinsic regulation of cytokinesis in EMS. (**A**) Schematic showing EMS and P2 cell-cell signaling during EMS cell fate specification. (**B**) Schematic showing how the EMS spindle angle was calculated. (**C**) Graph showing the EMS spindle angle (as a read-out for proper EMS cell fate specification) for different EMS (left) and P2 (right) cells within paired-blastomere doublets from control or

*Figure 8 continued on next page*

*Figure 8 continued*

*formin^cyk-1(ts)* embryos that successfully complete or fail in cytokinesis. Note: the doublets analyzed here are the same as those used in *Figure 7*. (D) Graph showing the frequency of successful cytokinesis for each cell type in intact *formin^cyk-1(ts); control(RNAi)* and *formin^cyk-1(ts); Src^src-1(RNAi)* embryos. Error bars, mean ± SD. Two tailed Mann-Whitney test (*Supplementary file 1*); n.s., no significance, p>0.05; *p≤0.05; **p≤0.01; ****p≤0.0001. (F) Model showing the role extrinsic and intrinsic factors in EMS and P2 cytokinesis. Temperature, 26°C; scale bar = 10 μm.

DOI: https://doi.org/10.7554/eLife.36204.025

The following figure supplements are available for figure 8:

**Figure supplement 1.** Disrupting Src^SRC-1 does not cause cytokinesis failure in control embryos.
DOI: https://doi.org/10.7554/eLife.36204.026

**Figure supplement 2.** Disrupting cellular polarity causes division failure in all cell types.
DOI: https://doi.org/10.7554/eLife.36204.027

from P2 cells. Together, our data demonstrate that both intrinsic and extrinsic mechanisms modulate cytokinesis in individual blastomeres in the absence of robust F-actin levels in the contractile ring.

## Discussion

Here, we used thermogenetics, drug treatment, and embryo microdissection to probe the mechanisms of cell-type-based variation in the regulation of cytokinesis among individual blastomeres within the four-cell *C. elegans* embryo. We found cell-type-specific differences in both the functional levels and temporal window of activity required for formin^CYK-1 activity, but not myosin-II^NMY-2 activity, during cytokinesis. We found that, similar to in the one-cell embryo (*Davies et al., 2014*), formin-^CYK-1 inhibition resulted in cytokinesis failure in ABa and ABp blastomeres. In contrast to the one-cell embryo and ABa/ABp cells, both EMS and P2 blastomeres were protected against cytokinesis failure and divided successfully with consistent frequency upon inhibition of formin^CYK-1 and in the absence of a robust F-actin contractile ring. This is not likely due to an F-actin-independent role for formin-^CYK-1, as we found cytokinesis in EMS and P2 was also more resistant to LatA, a pharmaceutical inhibitor of F-actin polymerization, than cytokinesis in ABa/ABp. Cell isolation and blastomere pairing experiments revealed that P2 is protected against cytokinesis failure due to cell-intrinsic regulation and independent of contact with other blastomeres. In contrast, we found that EMS is protected against cytokinesis failure by extrinsic regulation due to direct cell-cell contact with P2 and Src^SRC-1 mediated cell-fate signaling. This work establishes the early *C. elegans* four-cell embryo as a system to study cytokinetic diversity and suggests that, at least in the early *C. elegans* embryo, both cell-intrinsic and extrinsic mechanisms contribute to cell-type-specific cytokinetic diversity. This finding leads to three outstanding questions: First, how can cells divide in the absence of robust F-actin levels in the contractile ring? Second, what are the cell and molecular mechanism(s) that contribute to cytokinetic diversity? Third, what is the advantage of specifically protecting these cells against cytokinesis failure?

Although we could not detect enriched F-actin at the contractile ring in EMS or P2 in *formin^cyk-1(ts)* embryos at the restrictive temperature, we assume that a 'normal', although F-actin-poor, contractile ring still forms in the cells that divide successfully. In our hands, relative to the one-cell embryo, contractile ring F-actin levels are reduced at the four-cell stage and fluorescent signals from other cortical F-actin populations (such as the cell-cell junctions) make it much more challenging to specifically quantify F-actin levels at the contractile ring in individual cells within multicellular embryos. Thus, we assume that the contractile ring signal in EMS and P2 is simply too dim for us to detect over background signals from other cortical F-actin populations in *formin^cyk-1(ts)* mutants. In EMS and P2 cells that divide successfully upon formin^CYK-1 inactivation, contractile ring constriction progresses more slowly (*Figure 2—figure supplement 1*), suggesting that these cells are utilizing a sub-optimal contractile ring. Indeed, myosin^NMY-2 is still essential for cytokinesis in EMS and P2, indicating a 'normal' constricting actomyosin contractile ring is likely driving division in these blastomeres. Nonetheless, it is clear that, unlike one-cell embryos and ABa/ABp cells, EMS and P2 blastomeres are still somehow able to divide successfully with a substantially weakened F-actin

contractile ring, whether weakened by inhibition of formin[CYK-1] or application of low doses of Latrunculin A.

An obvious distinction of these more robustly dividing cells is that EMS and P2 divide asymmetrically, whereas ABa and ABp divide symmetrically (*Rose and Gönczy, 2014*; *Arata et al., 2010*). In the one-cell *C. elegans* embryo, we previously found that anterior-posterior cell polarity and the PAR proteins are essential for robust cytokinesis (*Jordan et al., 2016*). The PAR proteins localize to opposing cortical domains in P2 but are not obviously asymmetrically distributed in EMS (*Arata et al., 2010*). Little is known about cell polarity establishment and/or maintenance in EMS, but in P2, cell polarity is cell-intrinsically established, though proper orientation of polarity relative to the spindle is dependent on cell-extrinsic signaling from EMS (*Arata et al., 2010*). It is possible that cell polarity and the asymmetrically functioning PAR proteins (*Jordan et al., 2016*) or G-protein-coupled receptors, which have been implicated in cytokinesis in asymmetrically dividing *Drosophila* neuroblasts (*Cabernard et al., 2010*), could promote cytokinesis specifically in EMS and/or P2. Unfortunately, cell polarity in these later divisions is difficult to study, due to the lack of available conditional tools to disrupt cell polarity proteins specifically in four-cell embryos, while allowing normal cell polarity establishment and maintenance in asymmetrically dividing parental cells in the one- and two-cell embryos (i.e. fast-acting ts PAR mutants). While we found that whole embryo PAR protein disruption (by *par-6(RNAi)*) eliminates protection of cytokinesis in EMS and P2 (*Figure 8—figure supplement 2*), under these conditions, normal cell fate specification of the four-cell embryo is lost (*Bowerman et al., 1997*). Therefore, we cannot distinguish between a specific role for cell polarity during cytokinesis in EMS and P2 cells from a non-specific effect of completely changing the cell-fate patterning of the parental lineages.

One intriguing possibility is that other non-canonical (contractile ring-independent) mechanisms facilitate cytokinesis in EMS and/or P2 such as cytofission or polar cortical relaxation. In mammalian cultured cells, traction-mediated cytofission, driven by daughter cell elongation, crawling, and cell-substrate adhesion, can promote cell division in the absence of an actomyosin contractile ring (*Choudhary et al., 2013*; *Wheatley et al., 1997*). Cytofission is not likely to be the main driver of cytokinesis in these cells since, even upon blastomere isolation, we do not observe cell crawling behavior or increased daughter cell elongation in EMS or P2. Another possible driver of cytokinesis outside of the contractile ring is polar cortical relaxation, in which reduced cortical contractility outside of the division plane in the polar regions of the cell is proposed to facilitate actomyosin contractile ring constriction at the cell equator (*Wolpert, 2014*, *1960*; *Kunda et al., 2012*; *Rodrigues et al., 2015*; *Mangal et al., 2018*; *Lewellyn et al., 2010*). Polar cortical relaxation has been proposed to be regulated by Aurora A kinase activity, astral microtubules, and PP1-dependent phosphorylation of ezrin/radixin/moesin (ERM) family proteins (*Kunda et al., 2012*; *Rodrigues et al., 2015*; *Mangal et al., 2018*). Perhaps in EMS and/or P2, one or more of these cortical polar relaxation pathways outside of the contractile ring is upregulated to ensure robust cytokinesis in these blastomeres.

In EMS, we found that cytokinesis is extrinsically protected and this protection is correlated with cell-fate signaling from P2. Thus, it is possible that extrinsic cell fate signaling has direct downstream effects on the cytokinetic machinery in EMS. Both cell fate specification and asymmetric cell division in EMS are downstream of two partially redundant signaling pathways that influence similar downstream targets and depend on cell-cell contact between P2 and EMS: Wnt/Frizzled and Receptor Tyrosine Kinase (RTK)/Src signaling (*Goldstein, 1992*, *1993*, *1995a*, *1995b*; *Rocheleau et al., 1997*; *Thorpe et al., 1997*; *Bei et al., 2002*; *Arata et al., 2010*; *Schierenberg, 1987*; *Sugioka and Sawa, 2010*; *Berkowitz and Strome, 2000*). During this cell fate specification event, the Wnt[MOM-2] ligand from P2 activates Frizzled[MOM-5] receptors on the surface of EMS (*Rocheleau et al., 1997*; *Thorpe et al., 1997*; *Bei et al., 2002*; *Sugioka and Sawa, 2010*). In parallel, transmembrane-domain-containing RTK[MES-1] on P2 stimulates the same RTK[MES-1] on EMS in trans, promoting Src[SRC-1] activation in both cells (*Bei et al., 2002*; *Liu et al., 2010*; *Sugioka and Sawa, 2010*). Indeed, we found that both a high cell-cell contact area and proper spindle orientation, two indicators of successful cell fate signaling from P2 to EMS, are highly correlated with successful cytokinesis in EMS (*Figures 7* and *8*), and that depletion of Src[SRC-1] specifically causes division failure in the EMS cell, but not the P2 cell. In support of cell-fate signaling in cytokinetic diversity, both Wnt and Src-family kinase signaling have been implicated in both positive and negative regulations of cytokinesis in other cell contexts (*De Santis Puzzonia et al., 2016*; *Fumoto et al., 2012*; *Kasahara et al., 2007*;

*Soeda et al., 2013*; *Kamranvar et al., 2016*; *Avanzi et al., 2012*; *Sánchez-Bailón et al., 2012*; *Wu et al., 2014*; *Ikeuchi et al., 2016*; *Jungas et al., 2016*; *Kakae et al., 2017*), and in human cancer cell lines Wnt5a, its receptor Frizzled$^{Fz2}$, as well as a mediator of Wnt signaling, Dishevelled$^{Dvl2}$, all localize to the midbody at the division plane (*Fumoto et al., 2012*; *Kikuchi et al., 2010*). Thus, Wnt and/or Src cell fate signaling pathway components may themselves, or via their downstream targets, function to protect against cytokinesis failure in the absence of a robust actomyosin contractile ring.

Robust cytokinesis in P2 is cell-intrinsically regulated and independent of contact with other blastomeres within the four-cell embryo. The *C. elegans* P lineage forms the germline and cells within that lineage (including P2) inherit distinct levels of cellular organelles, cell polarity proteins, and transcriptional regulators compared to the other three blastomeres in the four-cell embryo. In flies, germ lineage cells also divide more robustly (by a specialized cytokinesis called cellularization) and are resistant to some loss of function mutations in Anillin that completely block cytokinesis in somatic lineage cells (*Field et al., 2005*). One possibility is that germline-specific inherited factors promote cytokinetic robustness in that lineage. For example, CAR-1 is an Sm-like protein essential for cytokinesis in the one-cell *C. elegans* embryo and is specifically inherited in the P lineage, including the P2 cell, via association with germline enriched non-membrane-bound, ribonucleoprotein (RNP) organelles called P-granules (*Audhya et al., 2005*; *Squirrell et al., 2006*). Perhaps the higher levels of CAR-1 (or of other P-granule associated factors) in P2 protect against cytokinesis failure in this blastomere.

What advantage is gained from protection of these specific cell lineages against cytokinesis failure upon formin$^{CYK-1}$ inactivation? In the early *C. elegans* embryo, six founder cells (AB, E, MS, C, D, and P4; *Figure 1A*) give rise to all cell lineages within the adult worm. In the four-cell embryo, EMS and P2 are upstream of founder cell formation within their lineages, while ABa and ABp are descendants of the first-born founder cell, AB (*Figure 1A*). We speculate that the EMS and P2 cells may be afforded extra levels of protection to ensure that they give rise to their founder cell descendants (E, MS, C, D, P4; *Figure 1A*), ensuring that all fates and lineages are represented in the developing worm. Additionally, both EMS and P2 undergo asymmetric cell divisions with each daughter cell inheriting specific organelles, transcriptional regulators, and other factors, so these blastomeres may undergo more selective pressure to develop protective mechanisms to ensure successful asymmetric cell division.

Together, our data show that both cell-intrinsic and extrinsic regulatory mechanisms contribute to cytokinetic diversity and promote robust cytokinesis when the contractile ring is weakened in specific cell types within the four-cell *C. elegans* embryo. Due to the similarities between cell fate signaling and cytokinetic machinery between worms and humans, we predict that similar regulation contributes to cytokinetic variation within cells of the human body to promote sensitivity or resistance to cytokinesis failure and therefore potentially mediate the onset or prevention of cell-type-specific pathologies.

## Materials and methods

### Strain maintenance

*C. elegans* were maintained on standard nematode growth media (NGM) plates seeded with OP50 *E. coli* as previously described (*Brenner, 1974*). Strain names and genotypes used in this study can be found in *Supplementary file 2*.

### Temperature control

Control and ts strains were maintained in an incubator (Binder) at a permissive temperature (16.0 ± 0.5°C). Live imaging was performed in a room with homeostatic temperature control set to the desired temperature at least 1 hr before the experiment. The temperature of the specimen was continuously monitored using at least three thermometers either attached to the objective or placed on the stage next to the sample.

## Rapid temperature shifts

Rapid temperature shifts were performed using a custom-built fluidic system called the Therminator (Bioptechs; [*Davies et al., 2014*]) with one water/isopropanol bath maintained at permissive temperature (16.0 ± 0.5°C) and a second bath at the restrictive temperature (26.0 ± 0.5°C).

## Embryo mounting and microdissection

For imaging intact embryos, young gravid hermaphrodites were dissected in 16°C M9 buffer (*Brenner, 1974*) and recovered embryos were mounted on a thin 2% agar pad as previously described (*Gönczy et al., 1999*). In *Figure 3*, *Figure 4—figure supplement 1*, and *Figure 5*, embryos were mounted with the 'hanging drop' method (*Davies et al., 2017*) using a SecureSeal spacer (Electron Microscopy Sciences, #70327–9S) to allow positioning of the embryo in the desired orientation.

For isolated blastomere imaging, young gravid hermaphrodites were dissected in ddH$_2$0, and then treated with alkaline hypochlorite to remove the eggshell. Embryos were placed in Shelton's growth medium [0.288 mg/mL inulin (Sigma Aldrich, #I2255), 2.88 mg/mL poly(vinylpolypyrrolidone) (Sigma Aldrich, #P0930), 0.0059x BME vitamins (Sigma Aldrich, #B6891), 0.0059x chemically defined lipid concentrate (ThermoFisher, #11905–031), 0.59x Penn-Strep (ThermoFisher, #15140–122), 0.52x *Drosophila* Schneider's Medium (ThermoFisher, #21720–024)], and 0.35x fetal bovine serum (ThermoFisher, 10438–018) (*Shelton and Bowerman, 1996*) before removal of the vitelline envelope and blastomere dissociation by repeated aspiration and ejection through a 30 µm needle (World Precision Instruments, #TIP30TW1) (*Klompstra et al., 2015*). In all cases, embryo isolation and blastomere dissections were performed at 16°C to allow development to the four-cell stage. For imaging, the isolated blastomeres were mounted in 20 µL of Shelton's growth medium in a Peltier-driven temperature-controlled chamber (26°C) (Oasis, Bioptechs, #15–160).

## Live cell imaging

Embryos were imaged using a spinning disc confocal unit (CSU-10; Yokogawa Electric Corporation) with Borealis (Spectral Applied Research) on an inverted microscope (Ti; Nikon) and a charge-coupled device camera (Orca-R2; Hamamatsu Photonics). Z-sectioning was done with a Piezo-driven motorized stage (Applied Scientific Instrumentation), and focus was maintained using Perfect Focus (Nikon) before each Z-series acquisition. An acousto-optic tunable filter was used to select the excitation light of two 100 mW lasers for excitation at 491 and 561 nm for GFP and mCherry, respectively (Spectral Applied Research), and a filter wheel was used for emission wavelength selection (Sutter Instruments). The system was controlled by MetaMorph software (Molecular Devices).

For most experiments, a 20 × 0.75 N.A. dry PlanApochromat objective was used, with 2 × 2 binning and 11 × 2 µm Z-sections collected every 60 s to measure cytokinetic progression, using embryos expressing GFP::PH and mCherry::H2B (*Audhya et al., 2005*) to label the plasma membrane and chromosomes respectively. In *Figure 1—figure supplement 1*, the F-actin markers PLST-1::GFP and LifeAct::RFP (*Ding et al., 2017*) were imaged with a 60 × 1.4 N.A. oil immersion PlanApochromat objective, with 2 × 2 binning and 4 × 0.5 µm Z-sections at the cortex. In *Figure 2—figure supplement 1*, a 60 × 1.4 N.A. oil immersion PlanApochromat objective, with 2 × 2 binning and 13 × 2.5 µm Z-sections collected every 30 s to measure cytokinetic progression with the GFP::PH and mCherry::H2B markers. In *Figure 3*, CYK-1::GFP was imaged at a single timepoint at the early stages of ingression using a 60 × 1.4 N.A. oil immersion PlanApochromat objective, with 2 × 2 binning and 35 × 1 µm Z-sections. In *Figure 4*, the F-actin markers, LifeAct::RFP and eGFP::Utrophin[ABD] (*Tse et al., 2012*) were imaged with a 60 × 1.4 N.A. oil immersion PlanApochromat objective, with 2 × 2 binning. In *Figure 4* and *Figure 4 – figure supplements 1* and *3*, 13 × 2 µm Z-sections were used. In *Figure 4 - figure supplement 2*, 65 × 0.5 µm Z-sections were used. In *Figure 5*, GFP::PH, mCherry::H2B, and FM 4–64 were imaged with a 40 × 0.95 N.A. dry PlanApochromat objective, with 2 × 2 binning and 13 × 2.5 µm Z-sections. In *Figures 6*, *7* and *8*, GFP::PH and mCherry::H2B in isolated blastomeres were imaged using a 40 × 0.95 N.A. dry PlanApochromat objective, with 2 × 2 binning and 9 × 2 µm Z-sections. In *Figure 6—figure supplement 1*, cells in intact embryos were imaged using a 40 × 1.25 N.A. water immersion Apochromat objective, with 1 × 1 binning and 51 × 1.0 µm Z-sections.

## Image analysis

MetaMorph (Molecular Devices) and ImageJ (National Institutes of Health [*Schneider et al., 2012*]) software were used for all data analyses. Cytokinetic phenotypes were scored using maximum projections of the Z-sections collected over the course of cell division. Cells were only scored if imaging began before anaphase onset and continued until the next cell cycle, when the daughter cell nuclei underwent anaphase and/or division failure. For the phenotypic analysis in *Figures 1* and *2*, cells were binned into five cytokinetic phenotypes; 'completes'=cell under observation divided successfully and the contractile ring remained closed when the daughter cell nuclei entered the next division cycle; 'fails, full constriction'=contractile ring constriction continues until no observable gap is seen with the PH::GFP membrane marker, but then regresses and cytokinesis fails; 'fails, partial constriction'=the cytokinetic furrow tip ingresses such that a double membrane forms (*Lewellyn et al., 2010*) but the contractile ring does not close before regressing; 'fails, weak constriction,' slight furrow ingression occurs, with weak membrane deformation, before regressing; 'fails, no constriction,' no furrowing or contractile ring ingression is observed following anaphase onset. Maximum intensity projections of GFP::PH and H2B::mCherry were used to monitor the time of anaphase onset and cytokinetic phenotype. In *Figure 2—figure supplement 1*, contractile ring diameter was measured by creating an X-Z projection so the whole contractile ring could be observed. For each time point, the Z-plane at which the ring diameter was widest was used for this measurement. Contractile ring diameter was plotted as a percentage of the initial diameter (at metaphase) over time. In *Figure 6—figure supplement 1*, the cell volume of isolated blastomeres was estimated by measuring the maximum cross-sectional area of each blastomere prior to anaphase (during mitosis), then calculating the radius and volume, assuming that the cells are spherical. Radius = $\sqrt{(\text{Area}/\pi)}$; Volume = $(4\pi r^3)/3$. The volume of cells in intact embryos was calculated as the sum of 23 or more 1 µm Z-section volumes, by measuring the cell area at each Z-plane. In *Figure 7E*, the area of contact between EMS and P2 was estimated assuming a circular contact interface and measuring the diameter of this interface in a maximum projection of multiple Z-sections, in the frame prior to anaphase onset in EMS. In *Figure 8C*, the EMS spindle angle was defined in the X-Y plane by drawing a line through the two chromosome masses in the first frame after anaphase onset in the EMS cell; this angle was measured relative to the long axis of the EMS-P2 doublet.

For the quantitative image analysis in *Figure 1—figure supplement 1*, F-actin levels in the one-cell embryo were analyzed using the maximum or sum intensity projection images of a four optical slice Z-section through the cell cortex. In *Figure 4*, F-actin levels in individual blastomeres in the four-cell embryo were analyzed from a Z-section through entire embryo using a maximum intensity projection (rather than a sum projection) to (1) increase the enriched cortical signal (where F-actin is localized), (2) reduce the effect of cell volume and position within the embryo on the measurements, and (3) to reduce the contribution of fluorescence signals analyzed from cytoplasm, cell-cell junctions, and adjacent cells. Line scans were 30.83 µm (P0 cells) 17.13 µm (EMS, ABa, and ABp cells) or 13.70 µm (P2 cells) long by 11.99 µm (P0 cells) or 3.42 µm (EMS, ABa, ABp, and P2 cells) wide, along the cell length perpendicular to the division plane. When cell polarity could be observed (in P0, EMS, and P2), the line was always oriented in the anterior to posterior direction. These line scans were normalized by subtracting the average of the initial and final 4.11 µm. Graphs showing the average fluorescence intensity at the division plane use the average value of the central 4.11 µm region of these line scans. Line scans were 13.70 µm long by 3.43 µm wide, along the cell length perpendicular to the division plane. When cell polarity could be observed (EMS and P2), the line was always oriented in the anterior to posterior direction. These line scans were normalized by subtracting the average value for a line scan adjacent to the embryo. Graphs showing the average fluorescence intensity at the division plane use the average value of the central 2.74 µm region of these line scans. In *Figure 3*, accumulation of formin[CYK-1]::eGFP was analyzed using a Z-series maximum intensity projection to select for the signal from the cortex next to the coverslip. In *Figure 3—figure supplement 3*, formin[CYK-1]::eGFP was analyzed using a Z-series sum intensity projection through the cell to measure the total levels of formin[CYK-1]::eGFP at the division plane (including cytoplasmic signal). In multicellular *C. elegans* embryos, we prefer maximum over sum intensity projection analysis to quantify fluorescence intensity differences, especially for cortically enriched proteins (e.g. formin[CYK-1]::eGFP, F-actin). Accurately comparing fluorescence intensity levels in the multicellular four-cell embryo is more challenging than in the one-cell embryo because: (1) individual cells within the four-

cell embryo vary in both their position within the four-cell embryo and in their axis orientation during cell division (ABa and ABp divide perpendicular to the long embryo axis whereas EMS and P2 divide parallel to this axis); (2) there is rotational variation of individual embryos relative to the coverslip in every image series; (3) individual cells within the four-cell embryo are of different volume and thus occupy different numbers of Z-sections within the image stack, (4) signals from adjacent cells contribute differently to the fluorescence intensity measured for each cell within the four-cell embryo, and (5) formin$^{CYK-1}$::eGFP is largely enriched at the cell cortex, but also occupies the cell cytoplasm. These issues impact sum projection analysis to a much greater extent than maximum projection analysis and in our opinion, render sum projection analysis less reliable for measuring the levels of formin$^{CYK-1}$::eGFP (or other cortical proteins) at the contractile ring in the four-cell embryo.

## CRISPR

Formin$^{CYK-1}$::eGFP expressing worms were generated using CRISPR/Cas9 to tag the endogenous *cyk-1* gene locus using a method described previously (*Dickinson et al., 2015*). The original self-excising cassette (SEC) repair plasmid pDD282 was modified, incorporating a 3' homology arm (570 bp of *cyk-1* coding sequence, with silent mutations to prevent gRNA/Cas9 cutting) and a 5' homology arm (690 bp of the 3' UTR of *cyk-1* and adjacent gene *rfl-1*) - for insertion of eGFP at the C-terminus of *cyk-1* (pJC340). pJC346 expresses the Cas9 protein, as well a gRNA, which was specifically modified to target *cyk-1* at the C-terminus (GCGAGAAGATCGTCTGTTGA<u>TGG</u> (PAM site underlined)) and was based on the plasmid pDD162. These were injected (pJC340, 10 ng/µL; pJC346, 55 ng/µL) into N2 young adults along with co-injection markers as described (*Dickinson et al., 2015*). Selection was performed as described (*Dickinson et al., 2015*) and rolling worms expressing formin$^{CYK-1}$::GFP were isolated. Homozygous formin$^{CYK-1}$::GFP integrants were heat-shocked for 5 hr at 32°C to remove the SEC (*Dickinson et al., 2015*). Successful integration and SEC excision were confirmed by PCR sequencing and visualization of formin$^{CYK-1}$::GFP on the spinning disc confocal microscope described above (see Live cell imaging). See *Supplementary file 2* for details of plasmids and sequences. Embryonic viability was scored by allowing individual hermaphrodites to lay eggs for ~24 hr at 20°C and each individual adult was transferred to a fresh plate each day for a total of ~72 hr. The number of progeny (embryos and larvae) were counted for each plate 24 hr after removal of the adult hermaphrodite.

## RNAi

Exonic sequences from the desired gene were cloned into the multiple cloning site of the L4440 vector using standard cloning techniques and then transformed into HT115 *E. coli* using CaCl$_2$ transformation as previously described (*Timmons et al., 2001*). RNAi primers and template DNA for each gene are listed in *Supplementary file 2*. RNAi feeding bacteria were grown in Luria broth with ampicillin (100 µg/mL) for 8–16 hr at 32°C. 300 µL of this culture was plated on RNAi plates (nematode growth media agar plates (*Brenner, 1974*) supplemented with 50 µg/mL ampicillin and 1 mM IPTG). These plates were allowed to dry and grow at 32°C for 24–48 hr. L1 stage worms were plated on RNAi plates and incubated at 16°C for 72 hr before dissection to obtain embryos.

## Eggshell permeabilization and LatruculinA treatment

L1 stage worms were plated on *perm-1(RNAi)* plates and incubated at 21°C for ~60 hr, to permeabilize the embryos by preventing normal eggshell formation (*Carvalho et al., 2011*; *Olson et al., 2012*). Young adult worms were picked directly into Shelton's growth medium (SGM, see *Embryo mounting and microdissection* above) and dissected to release early embryos. Four-cell stage embryos were then washed three times into SGM containing various concentrations of Latrunculin A (Tocris Bioscience, #3973, stored as a 2.5 mM stock in DMSO at −80°C), as well as 10 µM FM 4-64 (Thermo Fisher Scientific #T13320). Embryos were then transferred to a drop of the same growth medium on a No. 1.5 22 × 22 mm coverslip, and mounted with the 'hanging drop' method (*Davies et al., 2017*) using a SecureSeal spacer (Electron Microscopy Sciences, #70327–9S) to avoid changing the media composition or compressing the embryos. Successful eggshell permeablization was confirmed by the presence of the FM 4–64 dye on cell membranes in the first time point of the time lapse image series.

## Statistical analysis

Statistical significance was calculated in GraphPad Prism and Microsoft Excel. See also *Supplementary file 1* and the figure legends for a detailed description of statistical tests used for individual experiments.

## Acknowledgements

We thank all members of the Canman, Dumont, and Shirasu-Hiza labs for their support; Isaiah Thomas, Carrie Walsh, Amanda Smith, Nancy Quinn, Kania Rimu, Eva Sophia Blake, and Fallon Jung for lab assistance. We thank Diana Klompstra and Jeremy Nance (NYU) for invaluable guidance on *C. elegans* embryo micro-dissection; Iva Greenwald and Justin Benavidez (Columbia), Bob Goldstein (UNC), Daniel Dickinson (UT Austin); Morris Maduro (UC Riverside), Bruce Bowerman (U of Oregon), and Joseph and Jean Sanger (SUNY Upstate Med U) for advice and helpful discussions; Ronen Zaidel-Bar (Tel-Aviv U Med School) for worm strains; Shawn Jordan for assistance with ImageJ; Caroline Connors for assistance with cell fate analysis; and Sriram Sundaramoorthy, Yelena Zhuravlev, and (especially) Sophia Hirsch for critical comments on this manuscript. This work was funded by: a Charles H Revson Senior Fellowship in Biomedical Science (TD), (ANR-16-CE13-0020-01) (JD); FRM DEQ20160334869 (JD); NIH R01GM105775 (MSH); NIH R01AG045842 (MSH); NIH R01GM117407 (JCC); and NIH DP2OD008773 (JCC).

## Additional information

### Funding

| Funder | Grant reference number | Author |
|---|---|---|
| Charles H. Revson Foundation | Charles H. Revson Senior Fellowship in Biomedical Science | Tim Davies |
| Agence Nationale de la Recherche | ANR-16-CE13-0020-01 | Julien Dumont |
| National Institutes of Health | NIH R01GM105775 | Mimi Shirasu-Hiza |
| Fondation pour la Recherche Médicale | FRM DEQ20160334869 | Julien Dumont |
| National Institutes of Health | NIH R01AG045842 | Mimi Shirasu-Hiza |
| National Institutes of Health | NIH R01GM117407 | Julie C Canman |
| National Institutes of Health | NIH DP2OD008773 | Julie C Canman |

The funders had no role in study design, data collection and interpretation, or the decision to submit the work for publication.

### Author contributions

Tim Davies, Conceptualization, Data curation, Formal analysis, Supervision, Funding acquisition, Validation, Investigation, Visualization, Methodology, Writing—original draft, Project administration, Writing—review and editing; Han X Kim, Methodology, Made the CRISPR CYK-1::eGFP C. elegans strain in collaboration with Tim Davies; Natalia Romano Spica, Benjamin J Lesea-Pringle, Data curation, Formal analysis; Julien Dumont, Mimi Shirasu-Hiza, Conceptualization, Visualization, Writing—original draft, Writing—review and editing; Julie C Canman, Conceptualization, Resources, Data curation, Supervision, Funding acquisition, Visualization, Methodology, Writing—original draft, Project administration, Writing—review and editing

### Author ORCIDs

Tim Davies http://orcid.org/0000-0003-2247-9449
Julie C Canman http://orcid.org/0000-0001-8135-2072

Decision letter and Author response
Decision letter https://doi.org/10.7554/eLife.36204.032
Author response https://doi.org/10.7554/eLife.36204.033

## Additional files

### Supplementary files

• Supplementary file 1. A summary of the data analysis included in this manuscript, including number of experimental replicates and all statistical tests.
DOI: https://doi.org/10.7554/eLife.36204.028

• Supplementary file 2. A list of worm strains and RNAi constructs used in this study.
DOI: https://doi.org/10.7554/eLife.36204.029

• Transparent reporting form
DOI: https://doi.org/10.7554/eLife.36204.030

### Data availability

All data are included in the manuscript and supporting files. Due to their large size (100s of GBs), the source movies are available upon request to the corresponding author.

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
