## [Decision Letter]

[Editors’ note: a previous version of this study was rejected after peer review, but the authors submitted for reconsideration. The first decision letter after peer review is shown below.]

Thank you for submitting your work entitled "Cell-intrinsic and extrinsic control of cytokinetic plasticity" for consideration by *eLife*. Your article has been reviewed by three peer reviewers, and the evaluation has been overseen by a Reviewing Editor (Mohan Balasubramanian) and a Senior Editor. The reviewers have opted to remain anonymous.

Our decision has been reached after consultation between the reviewers. Based on these discussions and the individual reviews below, we regret to inform you that your work will not be considered further for publication in *eLife*.

The reviewers found the thermogenetics approach very interesting and also believe that the cell-type specific function of formin as novel and exciting. However, some major concerns were raised, which are reproduced verbatim below. In short, I see five key points that need consideration and addressing. I believe the suggested experiments may alter your conclusions significantly and may take well over two months to complete (our limit for revisions) and hence our decision to not consider it further.

However, if you are able to address the key points that I list below, we will be happy to consider a new manuscript from you. This we expect will move your story from being descriptive to mechanistic.

Main points to address: Many of these points have been raised by two or all of the reviewers.

1) Cyk1 ts mutant, whether it is defective for actin assembly at 15C, whether Cyk1 has an actin independent function. (raised by reviewer 2)..

2) Accuracy of quantitation approach of actin levels, use of other markers and other considerations raised by reviewers 2 and 3.

3) The strength of evidence on whether other formins are dispensable for cytokinesis in the 4 cell embryo (all reviewers).

4) Links between polarity proteins and cell type specific cytokinesis (raised by 2 and 3).

5) Links between Wnt/Src and cell type specific cytokinesis (reviewers 1 and 2).

Reviewer #1:

The process of physical separation of two cells, Cytokinesis, is regulated by numerous cell autonomous and non-autonomous cues. While constriction of an equatorial actomyosin network powers plasma membrane closure, the ability to regulate its spatiotemporally varies among cell types. In this work the authors present a thorough illustration of this principle in 4 cell stage of *C.elegans*. Using fast acting temperature sensitive alleles the authors show that cell fate influences the ability of a cell type to withstand Formin and actin perturbation. This is regulated both in a cell intrinsic and extrinsic fashion requiring neighbouring cell contacts.

The methodology and presentation of data is undoubtedly impressive. However, several conclusions appear to be an over-estimates the data. The text is well written yet offers little information on the process of cytokinesis and focuses almost exclusively on its outcome and failure. The authors on several occasions make categorical statements that at least to me appear inconclusive.

Major Points:

1) Figure 3-figure supplement 1

"This suggests that another formin-related protein is not compensating for loss of formin CYK-1 activity"

I concur that none of the formins at least individually show a strong synthetic defect with CYK-1ts. However that doesn't rule out the possibility that formins may act synergistically. Therefore, the conclusive statement doesn't stand merit.

2) "This suggests that EMS-P2 cell-cell contact mediates cell-extrinsic cues that protect against cytokinesis failure due to forminCYK-1 disruption in EMS".

The set of observations leading to this particular conclusion involves examining EMS-P2 pair after CYK-1 inactivation. Currently, it is not possible therefore to distinguish between the specific requirements of P2 versus requirement of an unspecified neighbouring cell. Whether the ABp-EMS pair would behave similarly cannot be hypothesised. Therefore the conclusion, I believe, is the presence of a neighbouring cell. Unless the authors prove or show specific mechanisms originating from P2, the current conclusion seems a stretch.

3) The Cell-extrinsic vs intrinsic angle indeed adds a dimension to this work. One would be curious to know, particularly in the EMS, whether the myosin-II ts yields an expected phenotype.

4) Figure 5.

I am unable to follow the logic here clearly. The spindle morphology has a direct impact on cytokinesis (beyond its spatial positioning). Spindles are therefore a cause of any cytokinesis defect under study. Proper spindle alignment (Figure 5E), does not guarantee successful cytokinesis. In CYK-1 ts, the distribution is visibly wider with some cases of proper spindle alignment and yet cytokinesis fails. I agree cases of successful cytokinesis seems to be correlative to proper spindle alignment. But this as a whole is incomplete and inconclusive. The authors do not make an attempt to conclude this section and one fails to see what exactly the authors allude to here.

Reviewer #2:

This manuscript reports the cell-type dependent phenotypes of a temperature-sensitive allele of cyk-1 formin in 4-cell stage *C. elegans* embryos. Inactivation of this f-actin nucleator during the first cell division by temperature upshift results in highly penetrant cytokinesis failure. Interestingly, while division of ABa and ABp cells was strongly inhibited by a similar temperature upshift during 4 cell stage, P2 and EMS divisions were more resistant and even at 26°C (the highest temperature that allows robust cell divisions in the wild type embryos) about 30-50% of these divisions completed successfully. Further, by blastomere isolation and recombination experiments, the authors revealed that cytokinesis of the EMS cell with reduced CYK-1 activity depends on the cell-cell contact between the EMS and P2 cells while the P2 division is independent of the contact with other cells. Based on these observations, the authors conclude that both the cell-intrinsic and extrinsic mechanisms protect against division failure due to defective contractile ring.

Establishment of an experimental system that allows further study of cell-type dependent variation in the mechanism of cytokinesis is a highly valuable achievement. However, there are major shortfalls in the current manuscript as below.

1) The authors implicitly assume that 1) the temperature sensitivity of cytokinesis failure is derived of the temperature sensitive inactivation of CYK-1 (ts) mutant protein and 2) the levels of residual CYK-1 activity after temperature upshit are invariable across the four different cell types (or completely inactivated in all the four cells). However, neither of these assumptions have been confirmed.

In Davies (2014), they compared CYK-1 FH1FH2C fragments with and without the L1015H mutation found in the cyk-1(ts) allele and showed that this mutation inactivates the in vitro actin polymerizing activity at 25°C. However, it remained unclear whether the CYK-1 mutant is active at 15°C or not (this actually should have been examined at the publication of the 2014 paper). Thus, currently, we can't exclude the possibility that the CYK-1 L1015H mutant protein is inactive at 15°C as well, and the mutant cell can complete cytokinesis at 15°C, but not at higher temperature, because of the presence of an unknown redundant pathway whose activity is intrinsically temperature-sensitive. If this was the case, the cell-type dependent ts phenotypes would not be reflecting the variable responses to the defective actin polymerisation but be indicating the variable activities of this redundant pathway.

Even if the temperature-sensitive cytokinesis failure of cyk-1(ts) was caused by the temperature-sensitive activity of the mutant protein, currently, there is no direct evidence that the cellular CYK-1 activity is inactivated uniformly across the four cell types. The cell type specificity might be caused by the variable level of the residual activity of CYK-1. If so, observed data should be interpreted as indicating the variations in a cellular mechanism for the expression of the CYK-1 activity, rather than the plasticity against the defective contractile ring.

2) The manuscript is rather descriptive and misses clear insight into the molecular mechanisms. In Jordan (2016), the authors reported the synthetic effects between the cyk-1(ts) and depletion of PAR proteins. Are the expression levels/cortical localization of the PAR proteins in the 4 cell types consistent with the cell type dependent phenotypes in cyk-1(ts) embryos? Does inactivation of the WNT or SRC signalling affect cytokinesis of the cyk-1(ts) mutant EMS cell?

3) Data about the f-actin levels in the cleavage furrow (Figure 3) are not convincing. About 50~70% of the cyk-1(ts) mutant EMS and P2 cells fail cytokinesis while 30~50% complete it. The successful cells might have more robust contractile ring than the unsuccessful ones. Successful cells and failed cells should be analyzed separately as the author did in Figure 5—figure supplement 1.

Reviewer #3:

The mechanisms of cytokinesis remain elusive despite decades of work. The lack of resolution of a universal mechanism may stem in part from the employment of a wide range of model cell types. It is unclear whether reported differences reflect cell-type specific distinctions or a general absence of redundant mechanisms in some systems. Davies and colleagues compared the molecular requirements for cytokinesis of several cells in the early *C. elegans* embryo. They combined the well-characterized fate specification of the cells with powerful, time-resolved perturbations of major conserved cytokinesis proteins via temperature sensitive mutant alleles. The manuscript is a technical tour-de-force, with exquisite cell manipulation experiments in addition to temperature shift work with intact embryos. It is exceptionally comprehensively referenced, and will set a new standard for the field. That said, since the authors are not able to resolve the problem of how certain cells divide following strong reduction of F-actin by CYK-1 (ts) upshift, I recommend a finite list of things to try and to consider. I feel confident that after these concerns are addressed, this manuscript will be suitable for publication in *eLife*.

Major Points:

1a) F-actin levels and distribution need to be quantified and presented for the AB lineage, comparing control and formin ts shifted embryos.

1b) The lack of an effect of targeting other formins is not fully satisfying, since no positive control is provided to ensure that they are depleted. One potential way to augment the exploration of these other formins is to consult a recent comparative transcriptomics study that may be able to verify which formins are expressed in the various early blastomeres (Tintori et al., Dev. Cell. 2016).

1c) Have the authors verified their analysis of F-actin levels using a second, distinct F-actin probe (LifeAct or the ABD of moesin)? It seems possible that these probes detect slightly different sub-populations of F-actin. I do not expect the authors to introduce fluorescently-labeled phalloidin or recombinant G-actin into embryos to distinguish different F-actin pools as has been done (see for example Burkel, von Dassow and Bement, Cytoskeleton 2007). The authors could label a non-formin-nucleated pool with a GFP-actin (Chai, Ou and colleagues, Nature Protocols, 2012).

1d) The authors should examine the localization of fluorescently-tagged CYK-1 during cytokinesis of the 4 cells they study.

2) In cases where cytokinesis is more successful than expected (insensitive cells), is initiation late? I.e. is overall duration longer than expected?

3) It would seem not outside the scope of this paper to combine the cyk-1 ts and a par mutant and shift the P2 after dissociation, to determine the role of polarity in protecting the P2.

4) Can the authors explain why higher temperatures do not always produce more severe phenotypic classification (i.e. Figure 1C, AB lineage, myosin and cyk-1 ts)? Is there compensation for the loss of function by an increase in things jiggling around? Similarly, why is shifting earlier not always worse (i.e. Figure 2B, EMS and P2, formin mutant)? Are un-scored embryos dying and scored, live ones less severely affected somehow?

[Editors’ note: what now follows is the decision letter after the authors submitted for further consideration.]

Thank you for resubmitting your work entitled "Cell-intrinsic and extrinsic mechanisms promote cell-type specific cytokinetic diversity" for further consideration at *eLife*. Your revised article has been favorably evaluated by Anna Akhmanova (Senior Editor), Mohan Balasubramanian (Reviewing Editor), and three reviewers.

The manuscript has been improved but there are some remaining issues that need to be addressed before acceptance, as outlined below:

The reviewers have raised a number of points and I have compiled the list of essential revisions below. The points raised do not require any additional experiments (unless you already have new data), but I would encourage you to state the conclusions without exaggeration and explain all of the points raised below satisfactorily.

In the discussion between reviewers, point 5 was emphasised as being important by ALL three reviewers.

1) Unfortunately, our suggestion of repeating an in vitro experiment to check the actin polymerization-promoting activity of the mutant version of CYK-1 at 16°C was neglected (or not presented). However, the authors measured the levels of F-actin in the contractile ring in the mutant embryos. Since the readout of this assay is the influence of the formin mutation on the complex cellular process of F-actin polymerization, it can also address our question though only indirectly. Anyway, irrespective of cell types and probes, the furrow F-actin levels in cyk-1(ts) mutant embryos at 16 °C were lower than those in wildtype embryos at 16 °C (Figure 4E, Figure 4—figure supplement 1C and Figure 4—figure supplement 3B). This suggests that CYK-1 L1015H mutant protein is not fully functional as the wildtype protein, as the authors admitted in the letter of rebuttal. However, although I am afraid that I might have just overlooked, I could not find any mention about this fact, which is important for the readers to understand the technical and conceptual limitation of the authors' approach, in the main text.

In the rebuttal, the authors wrote, "we have now measured the levels of F-actin in the contractile ring at both permissive and restrictive temperatures (16°C and 26°C) in control and formin cyk-1(ts) mutant embryos". However, I couldn't find any data for the ring F-actin levels in wildtype embryos at the restrictive temperature. Knowing now that CYK-4 ts mutant protein is not fully functional even at the permissive temperature, we can't exclude the possibility that the cytokinesis phenotype by acute temperature upshift might be triggered by acute inactivation of an unknown redundant mechanism for actin polymerization, which might be responsible for protecting the EMS and P2 cells from cytokines failure. In case of Figure 4E 'P2 cells', for example, drop of F-actin levels in cyk-1(ts) embryos from ~3 to ~0 by temperature upshift could be due to a) further inactivation of the mutant CYK-1 protein or b) inactivation of an unknown factor that contributes to the ring actin polymerisation. Data about F-actin levels in the wildtype embryos will be helpful for discriminating these possibilities. If the temperature upshift didn't affect the F-actin levels in the wildtype embryos or promoted it, this would provide a strong support for scenario a). On the contrary, if the temperature upshift drops the F-actin levels in the wildtype embryos from ~6 (16°C) to ~3 (26°C) or lower, it would be reasonable to conclude that the scenario b) is more likely.

2) I am afraid that I might be completely wrong, but I guess that the authors might already have the data of F-actin levels in wildtype embryos at 26°C since this is a very basic control. Whichever the results were, with the new data of the latrunculin A-sensitivity, the authors' key discovery in the current manuscript that the EMS and P2 cells are protected against cytokinesis failure due to the perturbed actin cytoskeleton will not be affected. Depending on the results, the authors might need to revise their basic assumption that the temperature shift causes a phenotype by acutely inactivating the mutant protein with a 'fast-acting ts mutation', on which they have been relying in previous publications. However, it is highly unnatural if the data of F-actin levels in wildtype embryos at 26°C is not shown. I strongly recommend showing the data of the F-actin levels in wildtype embryos at 26°C in at least one of Figure 4E, Figure 4—figure supplement 1C or Figure 4—figure supplement 3B (or Figure 1—figure supplement 1 although this is not really ideal), and properly discuss their implications on the possible mechanism for the acute induction of cytokinesis failure by temperature shift.

3) Figure 4—figure supplement 2C 'EMS'

A light blue half circle, probably derived from the markers of the graph points, is overlaid on a cartoon of a 6-cell stage embryo.

4) Figure 3—figure supplement 1

If we simply compare the control P2 cells (30 completion vs 40 failure, total n=70) with the inft-2(RNAi) (27 completion vs 14 failure, total n=41) by Fisher's exact test, the p-value will be 0.030. By Pearson's chi square test, it will be 0.032. This might be implying that inft-2, which is expressed at the 4-cell stage, might have an inhibitory role in the ring F-actin polymerization in the P2 cells although this is perfectly consistent with the authors' statement "we found that individual depletion of the other formin-related proteins did not decrease the frequency cytokinesis failure in any of the 4 blastomeres in formin cyk-1(ts) mutants." These simple calculations might not be appropriate for a complex dataset such as in Figure 3—figure supplement 1, to which care about multiple comparisons has to be paid, and the authors might have performed proper corrections, which might have made it >0.05, the significance level used in other figures. More details about the statistical method for interpretation of this valuable dataset should be provided.

5) The intensity of both F-actin and formin is calculated in various panels of Figure 1 and Figure 3 using maximum intensity projection. In my opinion, they should use sum intensity projection.

---

## [Author Response]

[Editors’ note: the author responses to the first round of peer review follow.]

Main points to address: Many of these points have been raised by two or all of the reviewers.1) Cyk1 ts mutant, whether it is defective for actin assembly at 15C, whether Cyk1 has an actin independent function. (raised by reviewer 2)..2) Accuracy of quantitation approach of actin levels, use of other markers and other considerations raised by reviewers 2 and 3.

To address reviewer points 1 and 2, we have now measured the levels of F-actin in the contractile ring at both permissive and restrictive temperatures (16ºC and 26ºC) in control and *formin^cyk-1^(ts)* mutant embryos using multiple fluorescently-tagged F-actin reporters (Lifeact::RFP, GFP::PLST-1, and GFP::Utrophin^ABD^). We found that contractile ring F-actin levels were lower in *formin^cyk-1^(ts)* mutant embryos than in control embryos at 16ºC (permissive temperature), and contractile ring F-actin levels were undetectable in *formin^cyk-1^(ts)* mutant embryos at 26ºC (restrictive temperature). This suggests the *formin^cyk-1^(ts)* mutant is indeed temperature sensitive for formin^CYK-1^-mediated F-actin polymerization at 26ºC and, despite providing sufficient formin^CYK-1^ activity to support organismal viability and germline function (Jordan et al., 2016), is not fully wildtype at 16ºC. To address these overall reviewer concerns in a different way, we also targeted F-actin polymerization independent of formin^CYK-1^ activity, by treating embryos with a pharmaceutical inhibitor of F-actin (Latrunculin A). These new results support our original hypothesis that the EMS and P2 cells are protected against cytokinesis failure when the actin cytoskeleton is weakened, and they are now included in this revised manuscript. For more details, please see also our responses to individual reviewer comments about both points below.

3) The strength of evidence on whether other formins are dispensable for cytokinesis in the 4 cell embryo (all reviewers).

To address this point, we have now added references about the expression of other formin-related genes and adapted the text as suggested by the reviewers. The diaphanous formin^CYK-1^ is the only formin shown to have a role in assembling contractile ring F-actin during cytokinesis in *C. elegans*. However, there are two other formin-related proteins expressed in 4-cell stage embryos: FRL-1 and INFT-2 (Hashimshony et al., 2015). Single cell RNAseq data from individual blastomeres of the 4-cell embryo revealed that none of the worm formin-related genes, including *cyk-1,* are significantly enriched in either EMS or P2, compared to in ABa or ABp cells (Tintori et al., 2016). Our mini-RNAi screen for additional formin-related proteins that might compensate for loss of formin^CYK-1^ activity was done at restrictive temperature (26ºC) in the *formin^cyk-1^(ts)* mutant background. We expected, that if another formin was functioning redundantly with formin^CYK-1^, then depletion of that formin would increase the rate of cytokinesis failure in EMS and/or P2. We did not find this to be the case, as depletion of the six *C. elegans* formin-family members did not increase the frequency of cytokinesis failure in any of the cells within the 4-cell embryo in *formin^cyk-1^(ts)* mutants. However, we agree that we cannot rule out insufficient protein depletion or that multiple formin-related proteins may work together to promote cytokinesis in these cells when formin^CYK-1^ activity is reduced and we now state this clearly in the manuscript text.

4) Links between polarity proteins and cell type specific cytokinesis (raised by 2 and 3).

We agree that cell polarity could be affecting cytokinesis in a cell type specific manner during asymmetric cell division. Both of the robustly dividing cells, EMS and P2 divide asymmetrically (Arata et al., 2010; Heppert et al., 2018), and in previous work we found that PAR proteins aid robust cytokinesis in the 1-cell embryo (Jordan et al., 2016). However, the exact mechanisms of polarity establishment and/or maintenance in EMS and P2 are not as well studied as in the 1-cell *C. elegans* embryo. The PAR proteins localize to opposing cortical domains in P2, but are not as obviously asymmetrically distributed in EMS (Arata et al., 2010). Cell polarity in these later divisions is difficult to study due to the lack of conditional tools that disrupt the cell polarity proteins specifically in 4-cell embryos, while allowing normal cell polarity establishment and maintenance in asymmetrically dividing 1- and 2-cell embryonic parental cells (*i.e.* fast-acting ts PAR mutants). Thus, to address this reviewer comment, we have now added data showing *par-6(RNAi)* eliminates the cell-type specific protection of EMS and P2 against cytokinesis failure in *formin^cyk-1^(ts)* mutant 4-cell embryos at restrictive temperature (Figure 8—figure supplement 1). We also note that in *par-6(RNAi)* treated embryos at the 4-cell stage, cell fate specification and lineage patterning have been massively disrupted due to the loss of polarity in the asymmetric 1- and 2-cell divisions.

5) Links between Wnt/Src and cell type specific cytokinesis (reviewers 1 and 2).

We have now examined cytokinesis in *formin^cyk-1^(ts)* mutant embryos at 26ºC following *control(RNAi)* or *Src^src-1^(RNAi)*. We found that cytokinesis in EMS is dependent on Src^SRC-1^ signaling, whereas cytokinesis in P2 is Src^SRC-1^ independent. These results are now included in the revised manuscript (Figure 8).

Reviewer #1:The process of physical separation of two cells, Cytokinesis, is regulated by numerous cell autonomous and non-autonomous cues. While constriction of an equatorial actomyosin network powers plasma membrane closure, the ability to regulate its spatiotemporally varies among cell types. In this work the authors present a thorough illustration of this principle in 4 cell stage of C.elegans. Using fast acting temperature sensitive alleles the authors show that cell fate influences the ability of a cell type to withstand Formin and actin perturbation. This is regulated both in a cell intrinsic and extrinsic fashion requiring neighbouring cell contacts.The methodology and presentation of data is undoubtedly impressive. However, several conclusions appear to be an over-estimates the data. The text is well written yet offers little information on the process of cytokinesis and focuses almost exclusively on its outcome and failure. The authors on several occasions make categorical statements that at least to me appear inconclusive.

We thank reviewer 1 for their constructive criticisms and we are pleased that they found the data and methodology in our original submission impressive. We have now added substantial new data and made changes to the text and figures based on their suggestions, and we are hopeful this reviewer will now find the manuscript suitable for publication.

Major Points:1) Figure 3—figure supplement 1"This suggests that another formin-related protein is not compensating for loss of formin CYK-1 activity"I concur that none of the formins at least individually show a strong synthetic defect with CYK-1ts. However that doesn't rule out the possibility that formins may act synergistically. Therefore, the conclusive statement doesn't stand merit.

We agree, and we have adjusted the text accordingly. It is true that our RNAi mini-screen results cannot exclude the possibility that multiple formin-related proteins function synergistically during cytokinesis in the absence of formin^CYK-1^ activity. The diaphanous formin^CYK-1^ is the only formin known to have a role in contractile ring F-actin assembly in *C. elegans*, and only two other formin-related proteins are expressed in the 4-cell embryo (*frl-1* and *inft-2;* (Hashimshony et al., 2015)), which could act redundantly when formin^CYK-1^ activity is reduced in EMS and/or P2 cells. Single cell RNAseq data of individual blastomeres of the 4-cell embryo indicates that no other formin-related protein is significantly enriched in either EMS or P2 compared with either the ABa or ABp cells (Tintori et al., 2016). Because we do not see a robust F-actin contractile ring during cytokinesis in EMS and P2 cells in *formin^cyk-^^1^(ts)* mutant embryos at restrictive temperature using fluorescently-tagged F-actin reporters, we argue it is unlikely another formin ‘takes over’ to promote the assembly of contractile ring F-actin when formin^CYK-1^ activity is reduced. Additionally, we have now perturbed actin polymerization independent of formin-disruption, by treating control, non-ts mutant, embryos treated with the pharmacological inhibitor of F-actin assembly, LatrunculinA (LatA). We found that, similar to what is observed in the *formin^cyk-1^(ts)* mutant, ABa and ABp blastomeres consistently failed at cytokinesis when treated with low doses of LatA (50, 67, or 80 nM), whereas the EMS and P2 blastomeres successfully completed cytokinesis at a high frequency at the same LatA concentrations. Both of these results suggest that a simple model of cell-specific redundancy between different formin family members is not the likely mechanism of robust cytokinesis in EMS and P2 cells in *formin^cyk-1^(ts)* mutant embryos. Nevertheless, we have added this possibility to this section of the Results.

2) "This suggests that EMS-P2 cell-cell contact mediates cell-extrinsic cues that protect against cytokinesis failure due to forminCYK-1 disruption in EMS".The set of observations leading to this particular conclusion involves examining EMS-P2 pair after CYK-1 inactivation. Currently, it is not possible therefore to distinguish between the specific requirements of P2 versus requirement of an unspecified neighbouring cell. Whether the ABp-EMS pair would behave similarly can not be hypothesised. Therefore the conclusion, I believe, is the presence of a neighbouring cell. Unless the authors prove or show specific mechanisms originating from P2, the current conclusion seems a stretch.

This is a great point and we have now taken both approaches suggested by the reviewer, and now include this critical control in the revised manuscript. We first assessed the frequency of cytokinesis success in isolated EMS blastomeres paired with ABd blastomeres from *formin^cyk-1^(ts)* mutants. In this experiment, ~50% of the time the cells are paired with ABa and ~50% of the time they are paired with ABp (after isolation, we term these cells ‘ABd’ for AB-daughter cells because they cannot be distinguished). We found that, similar to isolated (unpaired) blastomeres, EMS cannot undergo cytokinesis when in direct contact with ABd blastomeres. This suggests that a P2-specific signal is required to protect against cytokinesis failure in EMS when the contractile ring is weakened. These new data are included in the revised manuscript (Figure 7C).

To further address this reviewer concern, we also tested more directly if Src/Wnt cell fate signaling from P2 mediates protection against cytokinesis failure in EMS after depleting Src^SRC-1^ by RNAi. We found that following RNAi-mediated depletion of Src^SRC-1,^protection of EMS cytokinesis in intact 4-cell embryos from *formin^cyk-1^(ts)* mutants is lost, despite intact cell-cell contacts with the ABa, ABp and P2 cells. This result supports a model in which cytokinesis in EMS is dependent on activation of Src^SRC-1^ in EMS by direct contact with P2, as occurs during cell fate specification (Bei et al., 2002). These new data are included in the revised manuscript (Figure 8D).

3) The Cell-extrinsic vs intrinsic angle indeed adds a dimension to this work. One would be curious to know, particularly in the EMS, whether the myosin-II ts yields an expected phenotype.

We respectfully disagree that this experiment will substantially add to the manuscript, as we do not have evidence of cell-type specific myosin-II^NMY-2^ regulation during cytokinesis at the 4-cell stage. We found that in *myosin-II^nmy-2^(ts)* mutant embryos, all 4 blastomeres fail in cytokinesis equally at restrictive temperature (100% of the time). We expect upon isolation all 4 blastomeres would also fail in cytokinesis 100% of the time. While it is hypothetically possible that some cell-extrinsic pathway affects cytokinesis when myosin-II^NMY-2^ activity is reduced, we do not have any evidence to support a cell extrinsic pathway contributing to cytokinesis failure in a *myosin-II^nmy-2^(ts)* mutant. In the *formin^cyk-1^(ts)* mutant, we had in vivo evidence that there was a difference between cytokinesis in the cells in intact embryos versus in isolated blastomeres, as EMS and P2 were able to divide when formin^CYK-1^ activity was reduced in intact embryos, but only P2 could divide upon isolation. We do not have this evidence in *myosin-II^nmy-2^(ts)* mutants. Embryo microdissection and blastomere isolation experiments are laborious and not suited to exploratory work, unless a direct hypothesis is being tested. Thus, in our opinion this experiment is out of the realm of reasonable experimental requests.

4) Figure 5.I am unable to follow the logic here clearly. The spindle morphology has a direct impact on cytokinesis (beyond its spatial positioning). Spindles are therefore a cause of any cytokinesis defect under study. Proper spindle alignment (Figure 5E), does not guarantee successful cytokinesis. In CYK-1 ts, the distribution is visibly wider with some cases of proper spindle alignment and yet cytokinesis fails. I agree cases of successful cytokinesis seems to be correlative to proper spindle alignment. But this as a whole is incomplete and inconclusive. The authors do not make an attempt to conclude this section and one fails to see what exactly the authors allude to here.

We apologize for not stating the relationship between spindle angle and cell fate specification clearer in our original submission, or for in any way implying there an effect on overall spindle morphology. The spindle angle we measured is the angle of the EMS spindle relative to the P2-EMS cell-cell contact axis, not the spindle angle relative to the division plane (which are still perpendicular to each other, as occurs normally). To be clear, the axis of the cell division plane (and site of equatorial constriction) during cytokinesis is always perpendicular to the spindle angle in all EMS and P2 cells from control or *formin^cyk-1^(ts)* embryos in both intact and isolated blastomeres, as it is typically in animal cells undergoing cytokinesis. Thus, we do not believe mis-positioning the spindle angle relative to the division plane is affecting cytokinesis directly in this context and we do not see gross effect of formin^CYK-1^ disruption on spindle morphology.

The EMS spindle angle has been previously published as a read-out for correct P2 to EMS cell fate signaling in the *C. elegans* 4-cell embryo, and was the assay used to show that Wnt^MOM-2^ and Src^SRC-1^ signals act in parallel and non-redundantly for cell fate specification in EMS to promote cell division asymmetry and gut induction (Bei et al., 2002; Liu et al., 2010). However, we understand this is confusing and complicated. We have therefore significantly restructured this section to make the text clearer. We have now also added analysis showing that cytokinesis in EMS (within an intact embryo) is dependent on Src^SRC-1^ signaling, thus greatly improving the strength of our argument that cell fate signaling, which results in EMS spindle angle orientation, promotes robust cytokinesis in this blastomere.

It is also important to state that at this point of our studies we do not fully understand how P2contact dependent Src^SRC-1^ activation leads to protection against cytokinesis failure in EMS when the F-actin cytoskeleton is weakened with either the *formin^cyk-1^(ts)* mutant or with low doses of LatA. Thus, it is entirely possible that spindle signaling contributes to this cell type specific protection in EMS and/or P2. In light of this, we have now added this possibility to the discussion.

Reviewer #2:This manuscript reports the cell-type dependent phenotypes of a temperature-sensitive allele of cyk-1 formin in 4-cell stage C. elegans embryos. Inactivation of this f-actin nucleator during the first cell division by temperature upshift results in highly penetrant cytokinesis failure. Interestingly, while division of ABa and ABp cells was strongly inhibited by a similar temperature upshift during 4 cell stage, P2 and EMS divisions were more resistant and even at 26°C (the highest temperature that allows robust cell divisions in the wild type embryos) about 30-50% of these divisions completed successfully. Further, by blastomere isolation and recombination experiments, the authors revealed that cytokinesis of the EMS cell with reduced CYK-1 activity depends on the cell-cell contact between the EMS and P2 cells while the P2 division is independent of the contact with other cells. Based on these observations, the authors conclude that both the cell-intrinsic and extrinsic mechanisms protect against division failure due to defective contractile ring.Establishment of an experimental system that allows further study of cell-type dependent variation in the mechanism of cytokinesis is a highly valuable achievement. However, there are major shortfalls in the current manuscript as below.

We are pleased reviewer 2 found our establishment of the 4-cell *C. elegans* embryo for the study of cell-type variation in cytokinesis a valuable achievement. We also appreciate their helpful comments on our manuscript, especially their detailed reading of the Materials and methods and notice of several key errors. We hope that we have fully addressed their concerns.

1) The authors implicitly assume that 1) the temperature sensitivity of cytokinesis failure is derived of the temperature sensitive inactivation of CYK-1 (ts) mutant protein and 2) the levels of residual CYK-1 activity after temperature upshit are invariable across the four different cell types (or completely inactivated in all the four cells). However, neither of these assumptions have been confirmed.In Davies (2014), they compared CYK-1 FH1FH2C fragments with and without the L1015H mutation found in the cyk-1(ts) allele and showed that this mutation inactivates the in vitro actin polymerizing activity at 25°C. However, it remained unclear whether the CYK-1 mutant is active at 15°C or not (this actually should have been examined at the publication of the 2014 paper). Thus, currently, we can't exclude the possibility that the CYK-1 L1015H mutant protein is inactive at 15°C as well, and the mutant cell can complete cytokinesis at 15°C, but not at higher temperature, because of the presence of an unknown redundant pathway whose activity is intrinsically temperature-sensitive. If this was the case, the cell-type dependent ts phenotypes would not be reflecting the variable responses to the defective actin polymerisation but be indicating the variable activities of this redundant pathway.Even if the temperature-sensitive cytokinesis failure of cyk-1(ts) was caused by the temperature-sensitive activity of the mutant protein, currently, there is no direct evidence that the cellular CYK-1 activity is inactivated uniformly across the four cell types. The cell type specificity might be caused by the variable level of the residual activity of CYK-1. If so, observed data should be interpreted as indicating the variations in a cellular mechanism for the expression of the CYK-1 activity, rather than the plasticity against the defective contractile ring.

We agree that we did make these two assumptions in our initial manuscript. As discussed in our response to the reviewer and editorial comments above, we have now confirmed that the L1015H mutation in cyk-1(ts) causes loss of F-actin accumulation at the contractile ring, in a temperature-dependent manner. We initially tested this at the 1-cell stage to test the effect of temperature on *formin^cyk-1^(ts)* mutants for two main reasons: 1) the F-actin reporters are brighter and easier to image due to the absence of cell-cell contacts at the 1-cell stage; and 2) the essential role of formin^CYK-1^ at the 1-cell stage has been studied using non-ts mutations (Severson et al., 2002) and RNAi (Sonnichsen et al., 2005; Swan et al., 1998). These data are now included in the revised manuscript (Figure 1—figure supplement 1). We also measured contractile ring levels at the 4-cell stage in ABa, ABp, EMS, and P2 (Figure 4, and Figure 4—figure supplement 1-3).

We measured the levels of F-actin in the contractile ring at both permissive and restrictive temperatures (16ºC and 26ºC) in control and *formin^cyk-1^(ts)* mutant embryos using multiple fluorescently-tagged F-actin reporters (Lifeact::RFP, GFP::PLST-1, or GFP::Utrophin^ABD^). We found that contractile ring F-actin levels were lower in *formin^cyk-1^(ts)* mutant embryos than in control embryos at 16ºC (permissive temperature), and contractile ring F-actin levels were undetectable in *formin^cyk-1^(ts)* mutant embryos at 26ºC (restrictive temperature). This suggests the *formin^cyk-1^(ts)* mutant is indeed temperature sensitive for formin^CYK-1^-mediated F-actin polymerization at 26ºC and, despite providing sufficient formin^CYK-1^ activity to support organismal viability and germline function (Jordan et al., 2016), is not fully wildtype at 16ºC.

Thus, to address this reviewer concern in a different way, we also targeted F-actin polymerization independent of formin^CYK-1^ activity by treating embryos with a pharmaceutical inhibitor of F-actin (LatrunculinA or LatA). We found that, similar to what is observed in the *formin^cyk-1^(ts)* mutant, ABa and ABp blastomeres consistently failed at cytokinesis when treated with low doses of LatA (50, 67, or 80 nM), whereas the EMS and P2 blastomeres successfully completed cytokinesis at a high frequency at the same LatA concentrations. These new results support our original hypothesis that the EMS and P2 cells are protected against cytokinesis failure when the actin cytoskeleton is weakened, and they are now included in this revised manuscript (Figure 3).

2) The manuscript is rather descriptive and misses clear insight into the molecular mechanisms. In Jordan (2016), the authors reported the synthetic effects between the cyk-1(ts) and depletion of PAR proteins. Are the expression levels/cortical localization of the PAR proteins in the 4 cell types consistent with the cell type dependent phenotypes in cyk-1(ts) embryos? Does inactivation of the WNT or SRC signalling affect cytokinesis of the cyk-1(ts) mutant EMS cell?

To address this concern, we directly tested if Src/Wnt cell fate signaling from P2 mediates protection against cytokinesis failure in EMS by RNAi-inhibition of Src^SRC-1^. We found that following RNAi-mediated depletion of Src^SRC-1,^protection of EMS cytokinesis in intact 4-cell embryos from *formin^cyk-1^(ts)* mutants is lost, despite intact cell-cell contacts with the ABa, ABp and P2 cells. This result supports a model in which cytokinesis in EMS is dependent on activation of Src^SRC-1^ in EMS by direct contact with P2, as occurs during cell fate specification (Bei et al., 2002). These new data are included in the revised manuscript (Figure 8D).

We agree that cell polarity could be affecting cytokinesis in a cell type specific manner during asymmetric cell division. Both of the robustly dividing cells, EMS and P2, divide asymmetrically (Arata et al., 2010; Heppert et al., 2018), and in previous work we found that PAR polarity proteins aid robust cytokinesis in the 1-cell embryo (Jordan et al., 2016). However, the exact mechanisms of polarity establishment and/or maintenance in EMS and P2 are not as well studied as in the 1-cell *C. elegans* embryo. The PAR proteins localize to opposing cortical domains in P2, but are not as obviously asymmetrically distributed in EMS (Arata et al., 2010). Cell polarity in these later divisions is difficult to study due to the lack of conditional tools that disrupt the cell polarity proteins specifically in 4-cell embryos, while allowing normal cell polarity establishment and maintenance in asymmetrically dividing 1- and 2-cell embryonic parental cells (*i.e.* fast-acting ts PAR mutants). To address this reviewer comment, we have now added data showing *par-6(RNAi)* eliminates the cell-type specific protection of EMS and P2 against cytokinesis failure in *formin^cyk-1^(ts)* mutant 4-cell embryos at restrictive temperature (Figure 8—figure supplement 2). We also note that in *par-6(RNAi)* treated embryos at the 4-cell stage, cell fate specification and lineage patterning have been massively disrupted due to the loss of polarity in the asymmetric 1- and 2-cell divisions.

3) Data about the f-actin levels in the cleavage furrow (Figure 3) are not convincing. About 50~70% of the cyk-1(ts) mutant EMS and P2 cells fail cytokinesis while 30~50% complete it. The successful cells might have more robust contractile ring than the unsuccessful ones. Successful cells and failed cells should be analyzed separately as the author did in Figure 5—figure supplement 1.

We have now quantified contractile ring F-actin levels relative to the outcome of cytokinesis in *formin^cyk-1^(ts)* mutant strains expressing the RFP::Lifeact and PLST-1::GFP F-actin reporters. We compared contractile ring F-actin levels in EMS and P2 cells that divide successfully versus in those that fail to divide. Despite this effort, we have found no detectable levels of F-actin in the contractile ring in *formin^cyk-1^(ts)* mutant EMS or P2 blastomeres that succeed or fail to divide. Although we cannot detect it, it is possible that the successful cells have more robust contractile rings than the unsuccessful cells. However, even if this were true, this does not negate our results: EMS and P2 blastomeres are more successful at cytokinesis in the absence of formin^CYK-1^ activity or in the presence of LatA than the 1-cell embryo or ABa/ABp cells; this cytokinetic protection is intrinsically regulated for P2 cells and extrinsically regulated for EMS cells; and extrinsic regulation in EMS cells is dependent on Src^SRC-1^ signaling from P2 cells.

As we stated previously, all of these F-actin reporter strains are very dim and thus 1) are difficult to image and 2) an undetectable signal is difficult to interpret. We do not and cannot state that there is no F-actin in the contractile ring in these successfully dividing *formin^cyk-1^(ts)* mutant blastomeres, just that we cannot detect differences in F-actin levels between successful and unsuccessful EMS and P2 cells. The reason we present the contractile ring F-actin levels in *formin^cyk-1^(ts)* mutants is to demonstrate there is not a major redundant F-actin nucleating factor and/or massive F-actin stabilization in the EMS or P2 cells specifically, which could explain the ability of these two cells to divide in the absence of formin^CYK-1^ activity. We have made an effort to revise the text to ensure that we do not over-state this result.

Reviewer #3:The mechanisms of cytokinesis remain elusive despite decades of work. The lack of resolution of a universal mechanism may stem in part from the employment of a wide range of model cell types. It is unclear whether reported differences reflect cell-type specific distinctions or a general absence of redundant mechanisms in some systems. Davies and colleagues compared the molecular requirements for cytokinesis of several cells in the early C. elegans embryo. They combined the well-characterized fate specification of the cells with powerful, time-resolved perturbations of major conserved cytokinesis proteins via temperature sensitive mutant alleles. The manuscript is a technical tour-de-force, with exquisite cell manipulation experiments in addition to temperature shift work with intact embryos. It is exceptionally comprehensively referenced, and will set a new standard for the field. That said, since the authors are not able to resolve the problem of how certain cells divide following strong reduction of F-actin by CYK-1 (ts) upshift, I recommend a finite list of things to try and to consider. I feel confident that after these concerns are addressed, this manuscript will be suitable for publication in eLife.

We are pleased reviewer 3 found the data and methodology in our original submission impressive and thank them for their critical comments. We have now added new data and made substantial changes to the text based on their suggestions, and we are hopeful this reviewer will now find the manuscript suitable for publication.

Major Points:1a) F-actin levels and distribution need to be quantified and presented for the AB lineage, comparing control and formin ts shifted embryos.

We measured the levels of F-actin in the contractile ring in ABa and ABp at both permissive and restrictive temperatures (16ºC and 26ºC) in control and *formin^cyk-1^(ts)* mutant embryos expressing Lifeact::RFP, the brightest of the F-actin reporters at the 4-cell stage on our microscope system. We found that contractile ring F-actin levels were lower in *formin^cyk-1^(ts)* mutant embryos than in control embryos at 16ºC (permissive temperature) and contractile ring F-actin levels were undetectable in ABa and ABp (like in EMS and P2) in *formin^cyk-1^(ts)* mutant embryos at 26ºC (restrictive temperature). These results show that the *formin^cyk-1^(ts)* mutant is indeed temperature sensitive for formin^CYK-1^-mediated F-actin polymerization at 26ºC and, though not fully wildtype at 16ºC, provides sufficient formin^CYK-1^ activity to support actin polymerization, organismal viability, and germline function (Jordan et al., 2016).

1b) The lack of an effect of targeting other formins is not fully satisfying, since no positive control is provided to ensure that they are depleted. One potential way to augment the exploration of these other formins is to consult a recent comparative transcriptomics study that may be able to verify which formins are expressed in the various early blastomeres (Tintori et al., Dev. Cell. 2016).

We have now added information (and references) about the expression of other formin-related genes in 4-cell *C. elegans* embryos as suggested. The diaphanous formin^CYK-1^ is the only formin shown to have a role in assembling contractile ring F-actin during cytokinesis in *C. elegans*. However, there are two other formin-related proteins expressed in 4-cell stage embryos: FRL-1 and INFT-2 (Hashimshony et al., 2015). Single cell RNAseq data from individual blastomeres of the 4-cell embryo revealed that none of the worm formin-related genes, including *cyk-1,* are significantly enriched in either EMS or P2, compared to ABa or ABp cells (Tintori et al., 2016). Our mini-RNAi screen for additional formin-related proteins that might compensate for loss of formin^CYK-1^ activity was done at restrictive temperature (26ºC) in the *formin^cyk-1^(ts)* mutant background. We expected that if another formin was functioning redundantly with formin^CYK-1^, then depletion of that formin would increase the rate of cytokinesis failure in EMS and/or P2. We did not find this to be the case, as depletion of the six *C. elegans* formin-family members did not increase the frequency of cytokinesis failure in any of the cells within the 4-cell embryo in *formin^cyk-1^(ts)* mutants. However, we cannot rule out insufficient protein depletion or that multiple formin-related proteins may work together to promote cytokinesis in these cells when formin^CYK-1^ activity is reduced and we now state this clearly in the manuscript text.

To address this reviewer’s concern in a different way, we also targeted F-actin polymerization independent of formin^CYK-1^ activity by treating embryos with a pharmaceutical inhibitor of F-actin (LatrunculinA or LatA). We found that, similar to what is observed in the *formin^cyk-1^(ts)* mutant, ABa and ABp blastomeres consistently failed at cytokinesis when treated with low doses of LatA (50, 67, or 80 nM), whereas the EMS and P2 blastomeres successfully completed cytokinesis at a high frequency at the same LatA concentrations. These new results support our original hypothesis that the EMS and P2 cells are protected against cytokinesis failure when the actin cytoskeleton is weakened, and they are now included in this revised manuscript (Figure 3).

1c) Have the authors verified their analysis of F-actin levels using a second, distinct F-actin probe (LifeAct or the ABD of moesin)? It seems possible that these probes detect slightly different sub-populations of F-actin. I do not expect the authors to introduce fluorescently-labeled phalloidin or recombinant G-actin into embryos to distinguish different F-actin pools as has been done (see for example Burkel, von Dassow and Bement, Cytoskeleton 2007). The authors could label a non-formin-nucleated pool with a GFP-actin (Chai, Ou and colleagues, Nature Protocols, 2012).

We have now quantified contractile ring F-actin levels in the *formin^cyk-1^(ts)* mutant in strains expressing the F-actin reporters RFP::Lifeact, PLST::GFP, and GFP::Utrophin^ABD^. Using RFP::Lifeact, the brightest of the strains at the 4-cell stage on our microscope, as well as PLST::GFP, we also correlated the levels of F-actin in the contractile in ring in EMS and P2 cells that fail versus complete cytokinesis. Despite this effort, we found no detectable levels of F-actin in the contractile ring in either cells that complete or cells that fail to divide successfully. Although there is evidence to suggest different F-actin reporters label different sub-populations of F-actin (Belin et al., 2014; Bement et al., 2015), both of these markers are able to label the contractile ring in the 1-cell stage *C. elegans* embryo (Figure 1—figure supplement 1, RFP::Lifeact and PLST::GFP; Figure 4—figure supplement 3, eGFP::Utrophin^ABD^).

1d) The authors should examine the localization of fluorescently-tagged CYK-1 during cytokinesis of the 4 cells they study.

To address this concern, we used a CRISPR/Cas9 approach (Dickinson and Goldstein, 2016) to generate a fluorescently tagged formin^CYK-1^ (CYK-1::eGFP). We found that CYK1::eGFP localized nearly exclusively to the contractile ring in all 4 blastomeres, and contractile ring levels of CYK-1::eGFP in each cell did not significantly differ between the 4 cells. This is in agreement with single blastomere RNAseq analysis which showed no significant differences in *cyk-1* mRNA between the blastomeres (Swan et al., 1998; Tintori et al., 2016). These results are now included in Figure 5.

2) In cases where cytokinesis is more successful than expected (insensitive cells), is initiation late? I.e. is overall duration longer than expected?

To investigate this, we measured the rate of ingression in EMS and P2 cells in control and *formin^cyk-1^(ts)* mutant embryos at 26ºC and include this analysis in Figure 2—figure supplement 1. We observed that EMS and P2 cells in control embryos initiated and completed contractile ring constriction more quickly than in *formin^cyk-1^(ts)* mutant embryos, even when we limited our comparison to cells that divide successfully. When comparing the initiation time and initial rate of ingression, there was no clear difference between those that complete or those fail in cytokinesis. However, this analysis is obscured in cells that only weakly ingress due to the variable phenotype and by cell shape changes due to the movement and division of neighboring cells in the embryo. Due to the lack of ingression in ABa and ABp cells, we were unable to compare the initiation or ingression timing between the 4 cell types upon Formin^CYK-1^ inhibition.

3) It would seem not outside the scope of this paper to combine the cyk-1 ts and a par mutant and shift the P2 after dissociation, to determine the role of polarity in protecting the P2.

We agree that PAR-protein dependent cell polarity could be affecting cytokinesis in the P2 cells and were very keen to perform the experiment suggested by the reviewer (both in vivoand isolated P2 cells in vitro). Both of the robustly dividing cells, EMS and P2, divide asymmetrically (Arata et al., 2010; Heppert et al., 2018), and in previous work we found that PAR polarity proteins aid robust cytokinesis in the 1-cell embryo (Jordan et al., 2016). However, the exact mechanisms of polarity establishment and/or maintenance in EMS and P2 are not as well studied as in the 1-cell *C. elegans* embryo. The PAR proteins localize to opposing cortical domains in P2, but are not as obviously asymmetrically distributed in EMS (Arata et al., 2010). Cell polarity in these later divisions is difficult to study due to the lack of conditional tools that disrupt the cell polarity proteins specifically in 4-cell embryos, while allowing normal cell polarity establishment and maintenance in asymmetrically dividing 1- and 2-cell embryonic parental cells (*i.e.* fast-acting ts PAR mutants). Unfortunately, none of the ~5 available ts PAR alleles (Fievet et al., 2013) are “fast acting” (able to disrupt cell division asymmetry in the 1-cell embryo upon upshift <10-15 min before NEBD), and only one allele (*pkc-3(ne4250ts)*) disrupted cell polarity in the 1-cell embryo when upshifted >20 minutes before cell division, at least in our hands. Importantly, this one ts allele did not disrupt division asymmetry in 2- and 4-cell embryos (including in either EMS or P2). Thus, to address this reviewer comment, we have now added data showing *par-6(RNAi)* eliminates the cell-type specific protection of EMS and P2 against cytokinesis failure in *formin^cyk-1^(ts)* mutant 4-cell embryos at restrictive temperature (Figure 8—figure supplement 1). We also note that in *par-6(RNAi)* treated embryos at the 4-cell stage, cell fate specification and lineage patterning have been massively disrupted due to the loss of polarity in the asymmetric 1- and 2-cell divisions.

4) Can the authors explain why higher temperatures do not always produce more severe phenotypic classification (i.e. Figure 1C, AB lineage, myosin and cyk-1 ts)? Is there compensation for the loss of function by an increase in things jiggling around? Similarly, why is shifting earlier not always worse (i.e. Figure 2B, EMS and P2, formin mutant)? Are un-scored embryos dying and scored, live ones less severely affected somehow?

We agree that there are some intermediate temperatures and temporal upshifts where a higher temperature, or longer upshift, causes a weaker phenotype than a slightly higher temperature or longer upshift. We think this is unlikely to be due to cell death, as at these temperatures we don’t observe cell death in a temperature-dependent manner (we never see cells die upon upshift). We consistently follow cell division until the daughter cells begin to divide again, to ensure the cells are still cycling. Although cells in control embryos divide in a highly stereotypical manner, perturbation (*e.g.* by RNAi or with ts mutants) adds some variability cell biology in this system, and this variability seems to be increased at intermediate temperatures. Furthermore, the number of cells analyzed varied between the different temperatures and upshift times; thus a few blastomeres that fail or succeed in cytokinesis at a given temperature could have large effects depending on the size of the data set. To improve clarity, we have now added the n’s directly to all panels in Figures 1 and 2, with a further breakdown in the supplemental data file (Supplementary file 1).

[Editors' note: the author responses to the re-review follow.]

In the discussion between reviewers, point 5 was emphasised as being important by ALL three reviewers.1) Unfortunately, our suggestion of repeating an in vitro experiment to check the actin polymerization-promoting activity of the mutant version of CYK-1 at 16°C was neglected (or not presented). However, the authors measured the levels of F-actin in the contractile ring in the mutant embryos. Since the readout of this assay is the influence of the formin mutation on the complex cellular process of F-actin polymerization, it can also address our question though only indirectly. Anyway, irrespective of cell types and probes, the furrow F-actin levels in cyk-1(ts) mutant embryos at 16 °C were lower than those in wildtype embryos at 16 °C (Figure 4E, Figure 4—figure supplement 1C and Figure 4—figure supplement 3B). This suggests that CYK-1 L1015H mutant protein is not fully functional as the wildtype protein, as the authors admitted in the letter of rebuttal. However, although I am afraid that I might have just overlooked, I could not find any mention about this fact, which is important for the readers to understand the technical and conceptual limitation of the authors' approach, in the main text.

We apologize for not addressing this request to conduct pyrene actin assays with the CYK1(ts) mutant at 16ºC and 26ºC in our previous Resubmission; we did not understand that the reviewers were asking for this specifically. We understood that the reviewers were asking if the CYK-1(ts) mutant protein is fully functional for F-actin polymerization at permissive temperature, and we addressed this in vivoin our previous Resubmission. Unfortunately, it is simply not possible to do comparative in vitro activity studies at different temperatures with most biochemically purified ts mutant proteins. Our understanding is that ts mutant proteins show high protein instability at all temperatures once the ts protein is removed from the stabilizing environment of the cell, which contains molecular and structural protein chaperones. When we first published on this ts mutant, we asked Dr. David Kovar (U. Chicago), our collaborator who did the original pyrene actin assay comparing the in vitroactivity of wildtype CYK-1 to CYK-1(ts) (L1015H) (Davies et al., 2014), to do the pyrene actin assay at a low permissive temperature, despite the unlikeliness it would work. However, the Kovar lab did not have access to a cooling plate reader required for the assay and told us these are expensive and not commonly used. In any case, we agree with the reviewer: our in vivo data in this manuscript and in our previous publications on the *cyk-1(ts)* mutant, clearly show that the *cyk1(or596ts)* allele is not fully functional at permissive temperature. Importantly, though, CYK1(ts) mutant protein is functional enough at permissive temperature to support cell division, organismal development, and fertility. Moreover, when we examine the cytokinesis of single cells in *cyk-1(ts)* mutant embryos, cytokinesis is always successful at the permissive temperature in every cell-type we’ve analysed to date. These data were included in our previous submission and now this point is explicitly stated in the Results section of the main text:

“In *formin^cyk-1^(ts)* mutant 1-cell *C. elegans* embryos at 16ºC, F-actin is present in the contractile ring (although at lower levels than in control embryos), but upon upshift to 26ºC linear F-actin is no longer visible and cytokinesis fails (Figure 1—figure supplement 1 and Davies et al, 2014).”

In the rebuttal, the authors wrote, "we have now measured the levels of F-actin in the contractile ring at both permissive and restrictive temperatures (16°C and 26°C) in control and formin cyk-1(ts) mutant embryos". However, I couldn't find any data for the ring F-actin levels in wildtype embryos at the restrictive temperature. Knowing now that CYK-4 ts mutant protein is not fully functional even at the permissive temperature, we can't exclude the possibility that the cytokinesis phenotype by acute temperature upshift might be triggered by acute inactivation of an unknown redundant mechanism for actin polymerization, which might be responsible for protecting the EMS and P2 cells from cytokines failure. In case of Figure 4E 'P2 cells', for example, drop of F-actin levels in cyk-1(ts) embryos from ~3 to ~0 by temperature upshift could be due to a) further inactivation of the mutant CYK-1 protein or b) inactivation of an unknown factor that contributes to the ring actin polymerisation. Data about F-actin levels in the wildtype embryos will be helpful for discriminating these possibilities. If the temperature upshift didn't affect the F-actin levels in the wildtype embryos or promoted it, this would provide a strong support for scenario a). On the contrary, if the temperature upshift drops the F-actin levels in the wildtype embryos from ~6 (16°C) to ~3 (26°C) or lower, it would be reasonable to conclude that the scenario b) is more likely.

We now include data showing levels of contractile ring F-actin in control embryos at both 16ºC and 26ºC. We found the same levels of contractile ring f-actin in control embryos at both temperatures. As the reviewers suggest above, these data therefore rule out hypothesis (b) that an alternative temperature-sensitive mechanism promotes robust cytokinesis in the EMS and P2 cells. This result is also consistent with our acute LatA treatment results and suggests that cytokinesis in EMS and P2 is protected against failure when the contractile ring is weakened. These data can be found in the revised Figure 4 and Figure 4—figure supplement 3.

2) I am afraid that I might be completely wrong, but I guess that the authors might already have the data of F-actin levels in wildtype embryos at 26°C since this is a very basic control. Whichever the results were, with the new data of the latrunculin A-sensitivity, the authors' key discovery in the current manuscript that the EMS and P2 cells are protected against cytokinesis failure due to the perturbed actin cytoskeleton will not be affected. Depending on the results, the authors might need to revise their basic assumption that the temperature shift causes a phenotype by acutely inactivating the mutant protein with a 'fast-acting ts mutation', on which they have been relying in previous publications. However, it is highly unnatural if the data of F-actin levels in wildtype embryos at 26°C is not shown. I strongly recommend showing the data of the F-actin levels in wildtype embryos at 26°C in at least one of Figure 4E, Figure 4—figure supplement 1C or Figure 4—figure supplement 3B (or Figure 1—figure supplement 1 although this is not really ideal), and properly discuss their implications on the possible mechanism for the acute induction of cytokinesis failure by temperature shift.

Please see our response to point 2 above. We now include data showing contractile ring F-actin levels in control embryos at both 16ºC and 26ºC. We found the same levels of contractile ring f-actin in control embryos at both temperatures. These results are now included in the revised Figure 4 and Figure 4—figure supplement 3.

3) Figure 4—figure supplement 2C 'EMS'A light blue half circle, probably derived from the markers of the graph points, is overlaid on a cartoon of a 6-cell stage embryo.

Thank you for noticing; this has now been corrected.

4) Figure 3—figure supplement 1If we simply compare the control P2 cells (30 completion vs 40 failure, total n=70) with the inft-2(RNAi) (27 completion vs 14 failure, total n=41) by Fisher's exact test, the p-value will be 0.030. By Pearson's chi square test, it will be 0.032. This might be implying that inft-2, which is expressed at the 4-cell stage, might have an inhibitory role in the ring F-actin polymerization in the P2 cells although this is perfectly consistent with the authors' statement "we found that individual depletion of the other formin-related proteins did not decrease the frequency cytokinesis failure in any of the 4 blastomeres in formin cyk-1(ts) mutants." These simple calculations might not be appropriate for a complex dataset such as in Figure 3—figure supplement 1, to which care about multiple comparisons has to be paid, and the authors might have performed proper corrections, which might have made it >0.05, the significance level used in other figures. More details about the statistical method for interpretation of this valuable dataset should be provided.

We appreciate the time that the reviewers took to carefully examine our results. The hypothesis we tested –does another formin act redundantly with CYK-1 in EMS and/or P2? – is not supported by the data of this mini-screen. However, we agree that *inft-2(RNAi)* may have a partially suppressive effect on cytokinesis failure in *cyk-1(ts)* 4-cell embryos, and *inft-2* mRNA is reported to be expressed at the 4-cell stage (Tintori et al., 2016).

We analysed the cytokinesis outcome for each cell type, comparing between control and formin RNAi-treated embryos and testing for significance using Fisher’s exact test. Other than P2 cells in *inft-2(RNAi)*-treated embryos (as the reviewers noted), no comparison yielded a p-value less than or equal to 0.05. We agree that, for this experiment, correcting for multiple comparisons is important to prevent false positives. To our knowledge, while there is no standardised way of correcting for multiple comparisons when using Fisher's exact or Pearson's chi square tests, Bonferroni correction is frequently used. In Bonferroni correction, a (the value against which the p-value is compared) is divided by the number of comparisons. In this experiment, we used 6 pairwise comparisons between the control RNAi and each formin gene RNAi within a given cell type in the 4-cell embryo. Therefore, the Bonferroni correction adjusted value of a is 0.050/6 or 0.0083. None of the calculated p-values is smaller than 0.0083, indicating that none of the RNAi treatments had a significant effect on cytokinesis outcome. See also Supplemental file 1.

To clarify this in the manuscript, we added the following text to the figure legend for Figure 3—figure supplement 1):

“In *formin^cyk-1^(ts)* embryos from each individual formin RNAi treatment, the cytokinesis outcome for each cell type was compared with the cytokinesis outcome from the same cell type in *formin^cyk-1^(ts)* embryos treated with *control(RNAi)*, using Fisher’s exact text to examine the significance of any depletion. In all cases, the p-values were greater than 0.05, except for the P2 cell in *inft-2(RNAi); formin^cyk-1^(ts)* embryos which showed increased cytokinesis success with a p-value of 0.0299 relative to in *control(RNAi); formin^cyk-1^(ts)* embryos. However, using a Bonferroni correction to control for multiple comparisons adjusts α to 0.05/6 (or ~0.0083), suggesting that this result is not significant. See also Table S1.”

5) The intensity of both F-actin and formin is calculated in various panels of Figure 1 and Figure 3 using maximum intensity projection. In my opinion, they should use sum intensity projection.

We now include the sum intensity projection analysis in the revised Figure 1—figure supplement 1 for our analysis of F-actin levels in 1-cell *cyk-1(ts)* embryos at 16ºC and 26ºC. The results obtained with sum intensity projections were similar to those obtained with maximum intensity projections and are consistent with a temperature-sensitive F-actin polymerization defect in *cyk-1(ts)* embryos.

We also include the sum intensity analysis to Figure 3—figure supplement 1 in this Revision and similar to in Figure 1—figure supplement 1, the sum intensity analysis is consistent with the maximum intensity analysis and consistent with our overall hypothesis. However, we note here and in the text our concern with the sum intensity analysis for multi-cellular embryos. We include this analysis in response to the reviewers but would be happy to remove it if asked.

The main text now reads:

“Sum intensity projection analysis (total levels) revealed that formin^CYK-1^::eGFP is reduced in EMS and P2, versus ABa/ABp blastomeres (Figure 3—figure supplement 3). While the challenges of sum intensity projection analysis in multicellular embryos make it difficult to form conclusions about protein levels using this approach (see Materials and methods for additional information), these results are consistent with the maximum intensity projection analysis and do not support the hypothesis that resistance to formin^CYK-1^ inactivation in EMS and P2 is due to an endogenous enrichment of formin^CYK-1^ protein in these cells relative to ABa and ABp at the 4cell stage.”

We argue that the maximum intensity projection is a better way to quantify fluorescence intensity differences at the contractile ring between individual blastomeres for four reasons. The multicellular 4-cell embryo is much more challenging to image and compare levels between individual cells than is the 1-cell *C. elegans* embryo because: 1) individual cells within the 4cell embryo vary in both their position within the 4-cell embryo and in their orientation during cell division (ABa and ABp divide perpendicular to the long embryo axis whereas EMS and P2 divide parallel to this axis); 2) there is rotational variation of individual embryos relative to the coverslip in every time lapse image series; 3) individual cells within the 4-cell embryo are of different volume and thus occupy different numbers of z-sections within the image stack, and 4) CYK-1::GFP is largely enriched at the cell cortex, but also occupies the cell cytoplasm. These issues impact sum projection analysis to a much greater extent than maximum projection analysis and in our opinion, render sum projection analysis unreliable for measuring the levels of CYK-1::GFP at the contractile ring in the 4-cell embryo. In our sum projection analysis of contractile ring CYK-1::GFP levels in the 4-cell embryo, for example, the contractile ring signal scales roughly with cell volume, with smaller cells showing reduced signal than the larger cells. Again, this result is still consistent with our hypothesis that CYK-1 protein is not expressed at higher levels in EMS or P2, but we are not confident that this result indicates a real difference in CYK-1 levels in individual cells within the 4-cell embryo. We are happy to remove this analysis and retain our note in the Materials and methods explaining why if the reviewers and/or Editors allow this.

The Materials and methods section describing our analysis of contractile ring formin^CYK-1^::eGFP levels in individual cells within the 4-cell *C. elegans* embryo now reads:

“In Figure 3, accumulation of formin^CYK-1^::eGFP was analyzed using a Z-series maximum intensity projection to select for the signal from the cortex next to the coverslip. […] These issues impact sum projection analysis to a much greater extent than maximum projection analysis and in our opinion, render sum projection analysis less reliable for measuring the levels of formin^CYK-1^::eGFP (or other cortical proteins) at the contractile ring in the 4-cell embryo.”